# Active transcription and Orc1 drive chromatin association of the AAA+ ATPase Pch2 during meiotic G2/prophase

**Richard Cardoso da Silva**[1], **María Ascensión Villar-Fernández**[1,2], **Gerben Vader**[1]*

**1** Department of Mechanistic Cell Biology, Max Planck Institute of Molecular Physiology, Dortmund, Germany, **2** International Max Planck Research School (IMPRS) in Chemical and Molecular Biology, Max Planck Institute of Molecular Physiology, Dortmund, Germany

* gerben.vader@mpi-dortmund.mpg.de

**Data Availability Statement:** The ChIP-seq and RNA seq raw data employed in this study are deposited at the NCBI Gene Expression Omnibus

## Abstract

Pch2 is an AAA+ protein that controls DNA break formation, recombination and checkpoint signaling during meiotic G2/prophase. Chromosomal association of Pch2 is linked to these processes, and several factors influence the association of Pch2 to euchromatin and the specialized chromatin of the ribosomal (r)DNA array of budding yeast. Here, we describe a comprehensive mapping of Pch2 localization across the budding yeast genome during meiotic G2/prophase. Within non-rDNA chromatin, Pch2 associates with a subset of actively RNA Polymerase II (RNAPII)-dependent transcribed genes. Chromatin immunoprecipitation (ChIP)- and microscopy-based analysis reveals that active transcription is required for chromosomal recruitment of Pch2. Similar to what was previously established for association of Pch2 with rDNA chromatin, we find that Orc1, a component of the Origin Recognition Complex (ORC), is required for the association of Pch2 to these euchromatic, transcribed regions, revealing a broad connection between chromosomal association of Pch2 and Orc1/ORC function. Ectopic mitotic expression is insufficient to drive recruitment of Pch2, despite the presence of active transcription and Orc1/ORC in mitotic cells. This suggests meiosis-specific 'licensing' of Pch2 recruitment to sites of transcription, and accordingly, we find that the synaptonemal complex (SC) component Zip1 is required for the recruitment of Pch2 to transcription-associated binding regions. Interestingly, Pch2 binding patterns are distinct from meiotic axis enrichment sites (as defined by Red1, Hop1, and Rec8). Inactivating RNA-PII-dependent transcription/Orc1 does not lead to effects on the chromosomal abundance of Hop1, a known chromosomal client of Pch2, suggesting a complex relationship between SC formation, Pch2 recruitment and Hop1 chromosomal association. We thus report characteristics and dependencies for Pch2 recruitment to meiotic chromosomes, and reveal an unexpected link between Pch2, SC formation, chromatin and active transcription.

([http://www.ncbi.nlm.nih.gov/geo/](http://www.ncbi.nlm.nih.gov/geo/)), under accession nos. GSE138429 and GSE144835.

**Funding:** GV was funded by the European Research Council (ERC Starting Grant URDNA, agreement nr. 638197, [www.erc.europa.eu](www.erc.europa.eu)). RCS was funded by the CAPES-Humboldt fellowship from the Alexander von Humboldt Foundation (agreement nr. 99999.000021/2016-04, [www.humboldt-foundation.de](www.humboldt-foundation.de)). The funders had no role in study design, data collection and analysis, decision to publish, or preparation of the manuscript.

**Competing interests:** The authors have declared that no competing interests exist.

## Author summary

Meiosis is a specialized cellular division program that is required to produce haploid reproductive cells, also known as gametes. To allow meiosis to occur faithfully, several processes centred around DNA breakage and recombination are needed. Pch2, an AAA+ ATPase enzyme is important to coordinate several of these processes. Here, we analyze the genome-wide association of Pch2 to budding yeast meiotic chromosomes. Our results show that Pch2 is recruited to a subset of actively transcribed genes, and we find that active RNAPII transcription contributes to Pch2 chromosomal association. In addition, we reveal a general contribution of Orc1, a subunit of the ORC assembly, to Pch2 chromosomal recruitment. These findings thus reveal a connection between Pch2, Orc1 and RNAPII activity during meiosis.

## Introduction

Meiosis is a specialized developmental program dedicated to the production of genetically unique haploid gametes [1]. The production of haploid gametes is made possible by several meiosis-specific events, chief among them the event of homologous chromosome segregation during the first meiotic chromosome segregation event (*i.e.* meiosis I). Faithful segregation of homologs requires that initially unconnected homologous chromosomes are physically linked prior to segregation. Homolog linkage is achieved by interhomologue-directed crossover repair of programmed DNA double-strand breaks (DSBs) prior to meiosis I (*i.e.* during meiotic G2/prophase). DSBs are introduced by Spo11, a topoisomerase-like protein, which acts in conjunction with several accessory factors [2]. DSB formation happens in the context of a specialized, meiosis-specific chromosome architecture [3] [4]. Several protein factors, such as Hop1 and Red1 in budding yeast, (whose functional and structural homologs are HORMAD1/2 and SYCP2/3 in mammalians, respectively) [5] [6] [7] drive the assembly of chromosomes into linear arrays of chromatin loops that emanate from a proteinaceous structure termed the meiotic chromosome axis. Red1 and Hop1 co-localize with the meiotic cohesin complex (containing the meiosis-specific Rec8 kleisin subunit instead of the canonical Scc1) to form the molecular foundation of this typical meiotic 'axis-loop' chromosome structure [8, 9]. A zipper-like assembly called the synaptonemal complex (SC) polymerizes between synapsing homologous chromosomes [10], concomitantly with, and dependent on ongoing crossover repair of meiotic DSBs [11, 12]. In budding yeast, the Zip1 protein is an integral component of the SC, which is assembled onto the axial components of the loop-axis architecture [13, 14]. The SC likely acts as a signaling conduit that coordinates DSB activity and repair template preferences with chromosome synapsis [15–17]. A major role for the SC lies in directing the chromosomal recruitment of the hexameric AAA+ enzyme Pch2 [15, 18, 19], an important mediator of DSB activity, repair, and checkpoint function (reviewed in [20]). The molecular mechanisms of Pch2 recruitment to synapsed chromosomes remain poorly understood. In *zip1Δ* cells, Pch2 cannot be recruited to meiotic chromosomes (except to the nucleolus/rDNA; see below) [18]. However, this is unlikely via a direct molecular interaction. First, a specific Zip1-mutant (*zip1-4LA*) uncouples SC formation from Pch2 recruitment [15, 21]. Second, in cells lacking the histone H3 methyltransferase Dot1, Pch2 can be recruited to unsynapsed chromosomes in *zip1Δ* cells [22, 23]. Third, a recent report has linked topoisomerase II (Top2) function to Pch2 association with synapsed chromosomes [24], hinting at a connection between chromosome topology and Pch2 recruitment.

Functionally, the recruitment of Pch2 to synapsed chromosome is connected to the abundance of Hop1 on chromosomes [15, 18, 25]. The current model is that Pch2 recruitment to SC-forming chromosomal regions allows it to use its ATPase activity to dislodge Hop1 from synapsed regions [26–28], causing a coupling of SC formation to a reduction in DSB activity, interhomologue repair bias and checkpoint function [15, 20]. It is not clear whether Pch2 recruitment alone is sufficient to drive altered Hop1 dynamics upon synapsis, and additional aspects of chromosome metabolism, such as structural changes and post-translational modifications of axis factors [24, 29], have been implicated in crosstalk between Pch2 and Hop1.

In addition to its recruitment to euchromatic regions, Pch2 is recruited to the nucleolus, where it is involved in protecting specific regions of the ribosomal (r)DNA array (and rDNA-flanking euchromatic regions) against Spo11-directed DSB activity [18, 30]. The nucleolus is devoid of SC polymerization (and thus of Zip1), and nucleolar recruitment of Pch2 is dependent on Sir2 (a histone deacetylase) and Orc1 (a component of the Origin Recognition Complex (ORC)) [18, 30]. Strikingly, with the exception of Zip1, all factors that direct Pch2 recruitment (whether within the rDNA, or within euchromatin) are involved in chromatin function, be it modification (Dot1 and Sir2), binding (Orc1, via its bromo-adjacent homology (BAH) domain) or metabolism (Topoisomerase II). Together, these observations predict an intimate interplay between chromatin and Pch2 binding.

Inspired by this, and with the aim of increasing our understanding of Pch2 function on meiotic chromosomes, we generated a comprehensive map of Pch2 chromosomal association during meiotic G2/prophase. This analysis revealed specific binding sites of Pch2 across the genome. Within euchromatin, these sites map to regions of RNA Polymerase II (RNAPII)-driven transcriptional activity (*i.e.* a subset of active genes), and recruitment of Pch2 depended on active RNAPII-driven transcription. Interestingly, the Pch2 binding patterns identified here are distinct from meiotic axis enrichment sites (as defined by Red1, Hop1 and Rec8). Orc1 (and also other ORC subunits) are enriched at Pch2 binding sites, whereas no Pch2 can be found associated with origins of replication, which are the canonical binding sites of ORC [31]. Intriguingly, Orc1 inactivation triggers loss of Pch2 binding at active, euchromatic genes, demonstrating a connection between Pch2 and Orc1 that extends beyond their previously described shared rDNA-associated functions [30]. Although active transcription and Orc1 are equally present in meiotic and mitotic cells, we further show that ectopic expression of Pch2 in vegetatively growing cells is not sufficient to allow recruitment of Pch2 to the identified binding sites within actively transcribed genes. This suggests meiosis-specific requirements that license Pch2 recruitment. In agreement with this, we find that Zip1 is required for the recruitment of Pch2 to the identified transcription-associated binding regions.

Surprisingly, we find that interfering with the pool of Pch2 that associates with active RNAPII transcription does not lead to effects on the chromosomal association of Hop1, despite triggering a significant loss of Pch2 from meiotic chromosomes. This finding could indicate a more complex interplay between chromosome synapsis and Pch2 chromosomal recruitment and function than currently anticipated. We thus uncover characteristics and dependencies for Pch2 recruitment to meiotic chromosomes, and reveal a link between Pch2, SC formation, chromatin and active transcription.

## Results

We aimed to generate a detailed genome-wide mapping of the chromosomal localization pattern of Pch2, using chromatin immunoprecipitation followed by deep sequencing (ChIP-seq). For this, we employed an NH2-terminal 3xFLAG-tagged wild type version of Pch2 (Fig 1A

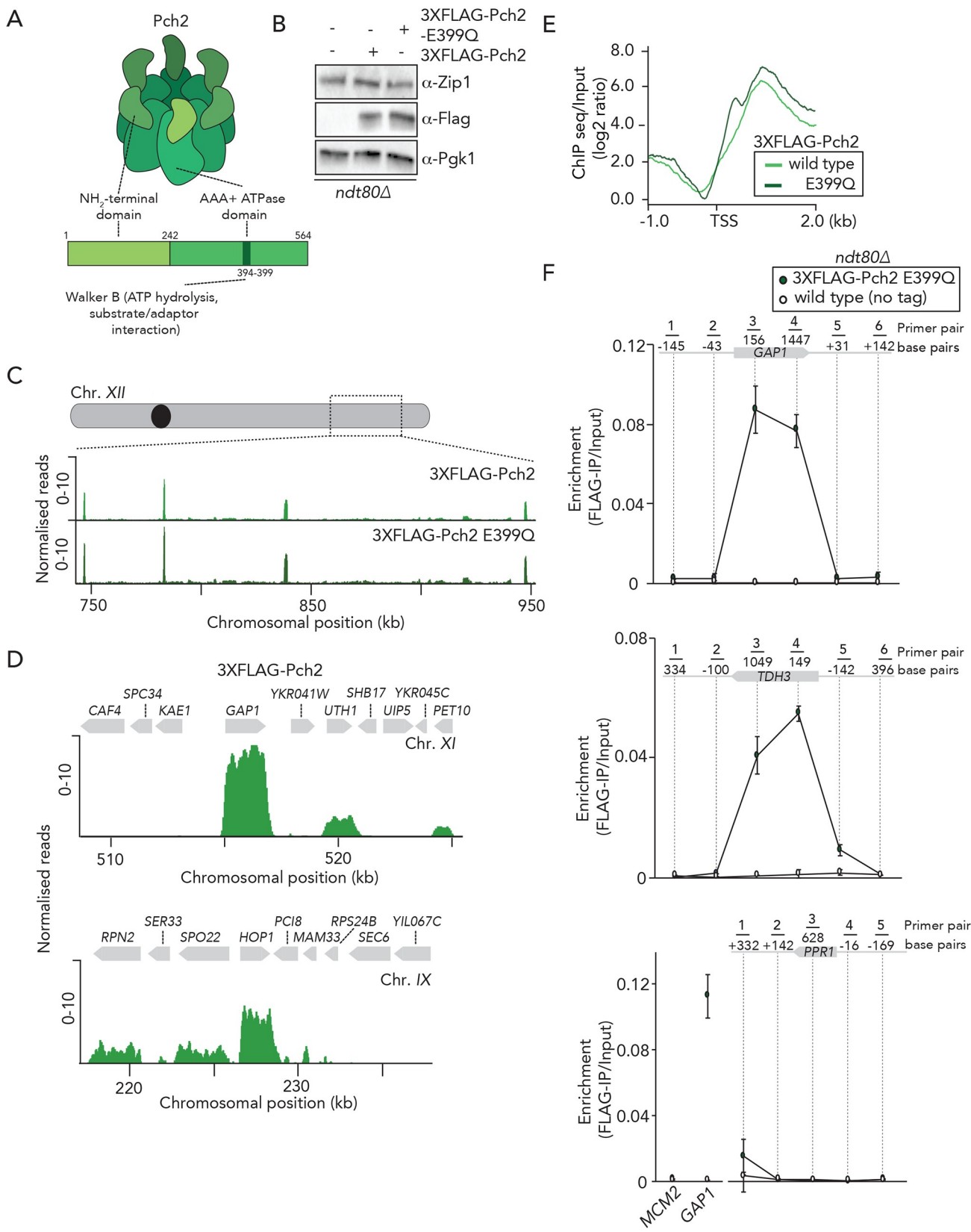

**Fig 1. Genome-wide analysis of Pch2 chromosome association.** A. Schematic of Pch2 domain organization. B. Western blot analysis of expression of Zip1 (upper panel) and 3XFLAG-Pch2 and 3XFLAG-Pch2-E399Q (lower panel) in *ndt80Δ* cells during meiotic G2/prophase at 4 hours after induction into the meiotic program. Pgk1 was used as a loading control. C. Genome browser view representative images (RPKM; see also Material and Methods) of ChIP-seq binding patterns for 3XFLAG-Pch2 and 3XFLAG-Pch2-E399Q. Shown is a region of Chromosome *XII* (chromosomal coordinates (kb) are indicated). D. High resolution Genome browser view representative images (RPKM; see also Material and Methods) of 3XFLAG-Pch2 binding patterns across two selected chromosomal regions (chromosomes *XI* and *IX*). Chromosomal coordinates and gene organization are indicated. E. 3XFLAG-Pch2 and 3XFLAG-Pch2-E399Q ChIP-seq enrichment normalized to inputs ($\log_2$). Datasets were aligned relative to the indicated positions covering Transcription Start Sites (TSS) and coding regions of 3XFLAG-Pch2 and 3XFLAG-Pch2-E399Q binding genes. F. ChIP-qPCR analysis tiling over the *GAP1/TDH3/PPR1* locus and downstream and upstream of their coding regions in *ndt80Δ* cells during meiotic G2/prophase (4 hours). Positions of the primers are indicated. Primer pairs for *GAP1*: 1) GV3197/G3198, 2) GV3195/GV3196, 3) GV2595/GV2596, 4) GV2599/GV2600; 5) GV3199/GV3200, 6: GV3201/GV3202. Primer pairs for *TDH3*: 1) GV3209/GV3210, 2) GV3207/GV3208, 3) GV2591/GV2592, 4) GV2593/GV2594; 5) GV3203/GV3204, 6: GV3205/GV3206. Primer pairs for *PPR1* (negative control): 1) GV3214/GV3215, 2) GV3211/GV3212, 3) GV2390/GV2391, 4) GV3216/GV3217; 5) GV3218/GV3219. *MCM2* (GV2392/GV2393) and *GAP1* (GV2595/GV2596) loci were used as a negative and positive control respectively. Error bars represent standard error of the mean of three biologically independent experiments performed in triplicate.

and 1B). This allele creates a functional protein: 3xFLAG-Pch2 was able to interact with Orc1 [30], and suppresses the synthetic spore viability defects of *rad17Δ pch2Δ* cells [32, 33] (S1A and S1B Fig). We also used a mutant allele harboring an E>Q substitution at position 399 within the AAA+ Walker-B motif (*pch2-E399Q*) (Fig 1A and 1B). This mutant is expected to impair ATP hydrolysis, and equivalent mutations have been used to stabilize interactions between AAA+ proteins and clients and/or adaptors [34]. We anticipated that Pch2-E399Q would exhibit increased association to chromosomal regions as compared to its wild type counterpart, which could aid in revealing details regarding Pch2 recruitment and/or function.

We investigated the progression of meiotic G2/prophase and chose to generate ChIP-seq datasets at 4 hours post-induction since at this time point our cultures showed a mixed population of cells in different phases of meiotic G2/prophase (as judged by SC polymerization status) during which Pch2 is known to play important roles (S1C Fig). Note that the strains that were used to generate ChIP-seq datasets (and the majority of subsequent experiments) were *ndt80Δ* in order to prevent exit from meiotic G2/prophase. We compared ChIP-seq datasets for wild-type and *E399Q* Pch2 (performed in triplicates in both cases) and found that these datasets exhibited highly correlated distributions, both at a genome-wide level and at individual loci S2A and S2B Fig and S3 Table (for number of peaks in different replicas). We plotted the pairwise correlation of normalized reads of shared Pch2 wild type and *E399Q* peaks (S2C Fig). The analyses indicate that the increased signal observed in *E399Q* binding relative to wild type is not originated from additional peaks exclusively detected in the Pch2-*E399Q* dataset (see also below).

We used both alleles (*i.e.* wild type and E399Q) for several follow-up experiments (see below). We called the peaks using MACS2 with a p-value of e10$^{-15}$ [9]. We found that ~98% of the peaks localized within the coding sequences (CDS) of a subset of RNAPII-transcribed genes distributed on all 16 budding yeast chromosomes (see S1 Table for a list of Pch2 binding CDS sites and Fig 1C and 1D for examples of typical binding patterns across selected chromosomal regions. See also S1D and S1E Fig for additional information and ChIP/input plots). Of note, the described peaks do not comprise those identified within the rDNA array on chromosome *XII* (see below). We did not observe Pch2-association within promoters (*i.e.* directly upstream of the transcriptional start sites (TSSs)) of these Pch2-bound genes (Fig 1D–1F). Pch2 peaks were evenly distributed throughout CDSs and located downstream of TSSs (Fig 1E). Association of Pch2 E399Q is stronger relative to wild-type Pch2, as judged by the differences in normalized read counts (S2D Fig and S1 Table). Based on the biochemical characteristics of AAA+ enzymes, the Pch2-E399Q is expected to exhibit stronger binding to clients and/or adaptors [34], and these increased binding patterns suggest that the observed binding sites represent biochemically meaningful interactions.

In addition to the observed association of Pch2 with RNAPII-transcribed genes, we also found Pch2-binding patterns within the rDNA array. Specifically, we found that Pch2 was associated with the *25S* RNAPI-transcribed locus, whereas we did not detect a significant association with *5S* (RNAPIII-transcribed) locus nor with intergenic regions (*NTS1* or *NTS2*) (data is available on http://www.ncbi.nlm.nih.gov/geo/), under accession no. GSE138429). These binding patterns might relate to the observed enrichment and function of Pch2 within the nucleolus [18, 30] and therefore warrant future investigation. In this manuscript, we however focus our attention on the Pch2 binding patterns across the non-rDNA, euchromatic part of the genome.

We performed a search for enriched Gene Ontology (GO) terms showing a representation for genes involved in various metabolic processes (S2 Table). In addition to these GO terms, Pch2-association was also enriched within certain sporulation-induced (*i.e.* meiosis-specific) genes (S1 Table). We next compared our Pch2 datasets to a genome-wide transcriptome (*i.e.* mRNA-seq) dataset that we generated from cells synchronously progressing through meiosis (this dataset was also generated in *ndt80Δ* cells 4 hours post induction of meiosis). We then plotted the normalized RNA-seq counts (TPM, transcripts per million) and assessed the expression levels of Pch2 binding genes following our criteria described in materials and methods. This showed that all genes occupied by Pch2 are transcribed during meiosis (S11C Fig), suggesting that transcription is involved in the recruitment of Pch2 to these CDSs. Processed RNA seq data can be found at http://www.ncbi.nlm.nih.gov/geo/), under accession no. GSE144835.

To investigate if transcriptional strength of defined genes was predictive of Pch2 binding, we stratified the transcribed genes from our RNA-seq dataset into high, medium and low expression strength (following previously established procedures [35]), and compared expression strength of Pch2-associated genes with these bins (S2E Fig). This analysis showed that Pch2-associated genes produce average mRNA levels, with a wide distribution. We detected only a weak correlation between the normalized reads score of individual Pch2-binding sites and the expression level of the corresponding CDS (Pearson's correlation, $R^2 = 0.3789$, S2F Fig). This indicates that, although Pch2 associates with actively transcribed genes, transcriptional strength *per se* likely plays, if any, only a minor role in dictating Pch2 binding. Underscoring this interpretation is the fact that many highly expressed genes do not show significant Pch2 enrichment peaks.

ChIP analysis can be plagued by artefactual ChIP-enrichments, which are mostly clustered at RNAPIII-transcribed genes, but some of which have also been observed to lie within highly-expressed RNAPII-transcribed genes [36]. We performed several analyses and experiments to exclude artefactual binding effects in our ChIP datasets, which we describe in detail in the Supplementary Data. Most importantly, we *i)* compared our datasets to reported artefactual binding sites [36] and found little overlap (S3A Fig); *ii)* did not detect binding of Pch2 to RNAPIII-transcribed tRNA genes (S3B Fig), contrary to what has been reported for artefactual ChIP-enrichments [36]; and *iii)* found that an inert nuclear protein (3xFLAG-dCas9) did not show binding to a defined Pch2-associated site (as would be expected for artefactual ChIP signals), as tested by ChIP-qPCR (S3C and S3D Fig). Based on these and additional experiments that are described below and in the Supplementary Data, we are confident that our Pch2 datasets inform on physiologically relevant biological behavior.

Based on our ChIP-seq results, we employed ChIP followed by real-time quantitative PCR (ChIP-qPCR) to explore the connection between Pch2 and transcription. We designed oligos that tile over the CDSs and upstream and downstream regions of two Pch2-bound genes: *GAP1* and *TDH3*. *PPR1*, an RNAPII-transcribed gene to which Pch2 showed no association by ChIP-seq was used as a negative control. We confirmed transcriptional activity at *GAP1* and

*PPR1* by ChIP analysis of active RNAPII (via ChIP of α−PolII-phospho-Ser5, which is used as a read-out of active engagement of RNAPII during transcription elongation (reviewed in [37] (S2H Fig)). 3xFLAG-Pch2-*E399Q* associated with the CDSs of both *GAP1* and *TDH3*, whereas no significant association was observed at the *PPR1* locus (Fig 1F and S2G Fig, see also S3 Table for a spreadsheet containing, in separate sheets, the underlying numerical data for figure panels containing ChIP-qPCR data). Binding patterns at *GAP1* and *TDH3* closely mirrored the narrow CDS-specific patterns that we found in our ChIP-seq analysis (for example, compare signals for *GAP1* in Fig 1D and 1F). We validated the association of wild-type Pch2 and Pch2-E399Q to two additional selected binding genes (*HOP1* and *SSA1*) by ChIP-qPCR, and we confirmed increased binding of Pch2-E399Q as compared to wild-type Pch2 (S2I and S2J Fig). In addition to its catalytic AAA+ domain, Pch2 also possess a non-catalytic NH2-terminal domain (NTD) (Fig 1A) [20]. The NTDs of AAA+ ATPases are required to allow AAA + proteins to interact with clients and adaptors [26] [38] [20, 34]. Removal of the NTD of Pch2 abrogated the association of Pch2 to individual selected genes, indicating that the NTD is required for recruitment of Pch2 to gene bodies (S2I and S2J Fig). In this regard, it is important to note that the expression of Pch2 lacking its NTD (Pch2 243–564) was significantly lower as compared to wild type Pch2 (see also S4 Table for a spreadsheet containing, in separate sheets, selected western blot signals for representative western blot images presented in the manuscript). In addition, it has been reported that the NTD of Pch2 harbors a nuclear localization signal [39]. As such, Pch2 243–564 is likely inefficiently localized to the nucleus [39]. In conclusion, we find that during meiotic G2/prophase, Pch2 associates within the body of a selected group of RNAPII-associated genes, and that recruitment depends on characteristics of AAA+ proteins.

Hop1, a HORMA-domain containing client of Pch2 is a central component of the meiotic axis structure [8, 15, 18, 25–28, 40]. Zip1-dependent SC assembly (which drives Pch2 recruitment [18]), is established on the axial element of the meiotic chromosome structure, and Hop1 and Zip1 are therefore expected to reside in molecular proximity of each other (at and near chromosome axis sites, respectively). As such, one hypothesis is that Pch2 is also enriched at meiotic axis-proximal sites, where it might be acting on its client, Hop1. To investigate whether our ChIP dataset could inform on this idea, we compared the binding patterns of Pch2 to those of axial components (*i.e.* Red1, Hop1 and Rec8) with available ChIP-seq datasets [9] (Fig 2A). Hop1, Red1 and Rec8 showed highly similar binding patterns [8] [9], but the binding patterns of both wild-type and ATP-hydrolysis deficient Pch2 qualitatively diverged from the patterns of these axial elements: Pch2 patterns did not show the similar frequency along chromosomal regions, and, on genome-wide level, showed little overlap with the binding patterns of meiotic axis-factors (Fig 2A and 2B and S4A and S4B Fig). We propose that, within the loop-axis organization of meiotic chromosomes, Pch2 associates with (a selected group of) genes located within loops that are located away from the Hop1-Red1-Rec8-defined axis (Fig 2C).

We next investigated the effect of RNAPII transcriptional activity on Pch2 occupancy on meiotic chromosomes. To inhibit RNAPII-dependent transcription, we initially used 1,10-Phenanthroline – a small molecule which has previously been described to inhibit RNAPII-dependent transcription [41]. Meiotic yeast cultures expressing 3XFLAG-Pch2 E399Q were treated with 1,10-Phenanthroline for 1 hour (S5A Fig). Under these conditions, we observed a substantial effect on *GAP1* mRNA levels, in cells treated with 1,10-Phenanthroline, whereas Pch2 protein levels were unaffected (S5B and S5C Fig). Inhibition of global transcription reduced Pch2 association to *GAP1* gene by ~50% compared to the mock-treated control situation (S5D Fig), consistent with a role for transcription in promoting the recruitment/association of Pch2 to regions of active transcription.

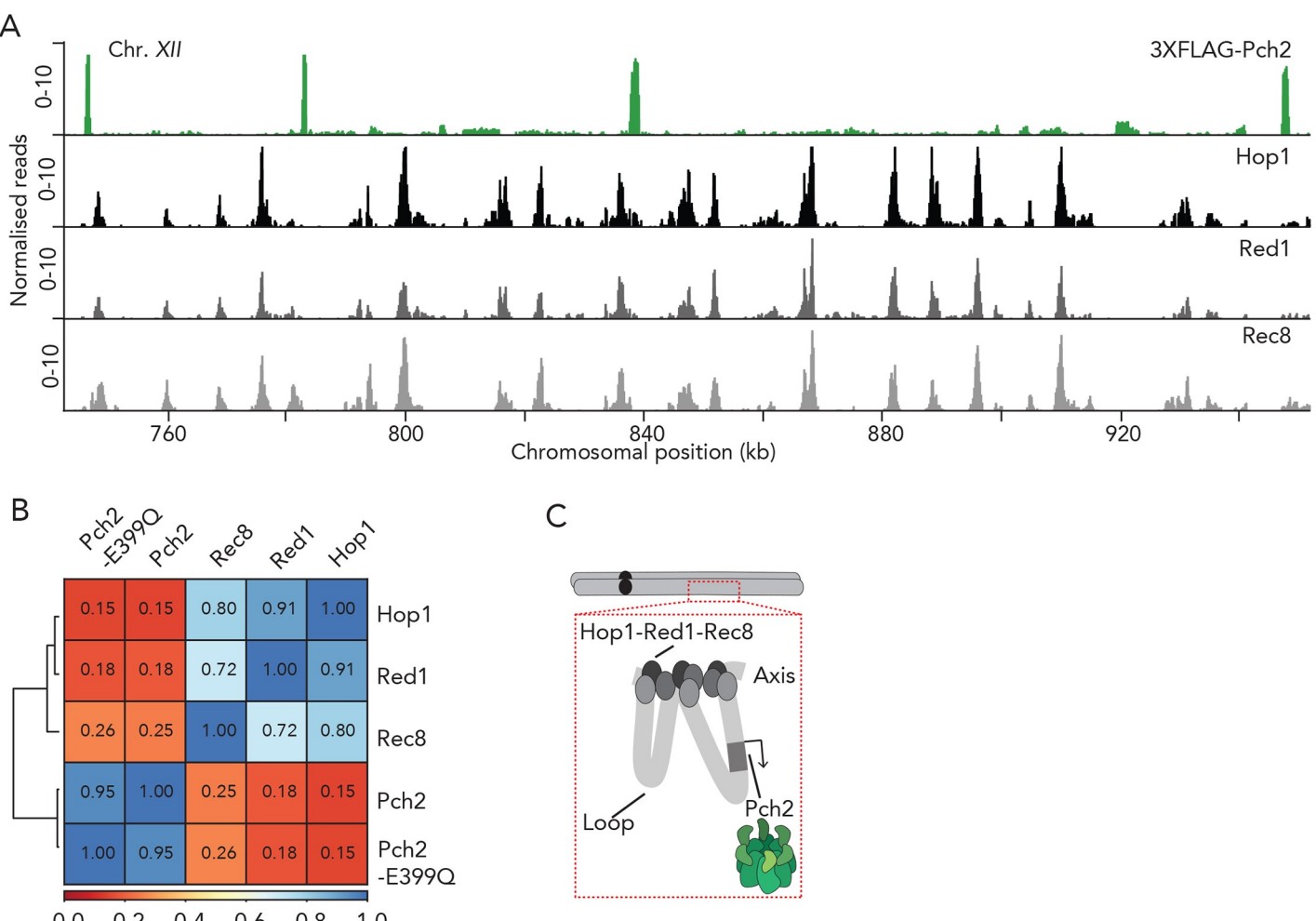

**Fig 2. Pch2 and meiotic axis sites.** A. Genome browser view representative images (RPKM; see also Material and Methods) of ChIP-seq binding patterns for 3XFLAG-Pch2, Hop1, Red1 and Rec8. Data for Hop1, Red1 and Rec8 are from [9]. Shown is a region of Chromosome *XII* (chromosomal coordinates (kb) are indicated). B. Hierarchically clustered heatmap based on correlation coefficients using from 3XFLAG-Pch2-E399Q ChIP-seq datasets as inputs. Hop1, Red1 and Rec8 ChIP-Seq datasets are from [9]. Spearman's correlation values are indicated. C. Model depicting the proposed localization pattern of Pch2 on loops, within the meiotic chromosome loop-axis structure.

To achieve a more complete and specific inhibition of RNAPII, we employed the anchor-away technique [42], which has been used to successfully deplete chromosomal proteins during meiosis [15, 24, 43, 44]. This technique is based on an inducible dimerization system that rapidly depletes nuclear proteins based on ribosomal flux, with the aid of a tagged anchor protein, Rpl13A (Rpl13a-2XFKBP12). Rapamycin induces the formation of a ternary complex with a protein of interest that is tagged with FRB (FKBP12-Rapamycin Binding-FRB domain of human mTOR) (Fig 3A). Successful anchor-away-based inhibition of RNAPII has been described in vegetative cells [45], and we similarly tagged the largest subunit of *RNAPII (Rpo21)* with the FRB tag (Fig 3A and S5E Fig). As expected, *rpo21-FRB* cells exhibited severe growth defects in the presence of rapamycin (Fig 3B). Immunofluorescence of meiotic chromosome spreads after exposure with Rapamycin for 30 minutes demonstrated efficient nuclear depletion of Rpo21-FRB during meiosis (Fig 3C and 3D). We note that Rpo21 was localized to meiotic chromosomes, in a dotted pattern that did not show strong similarity with Zip1 or Hop1 binding patterns (Fig 3D and S5F Fig). We performed ChIP-qPCR analysis, using

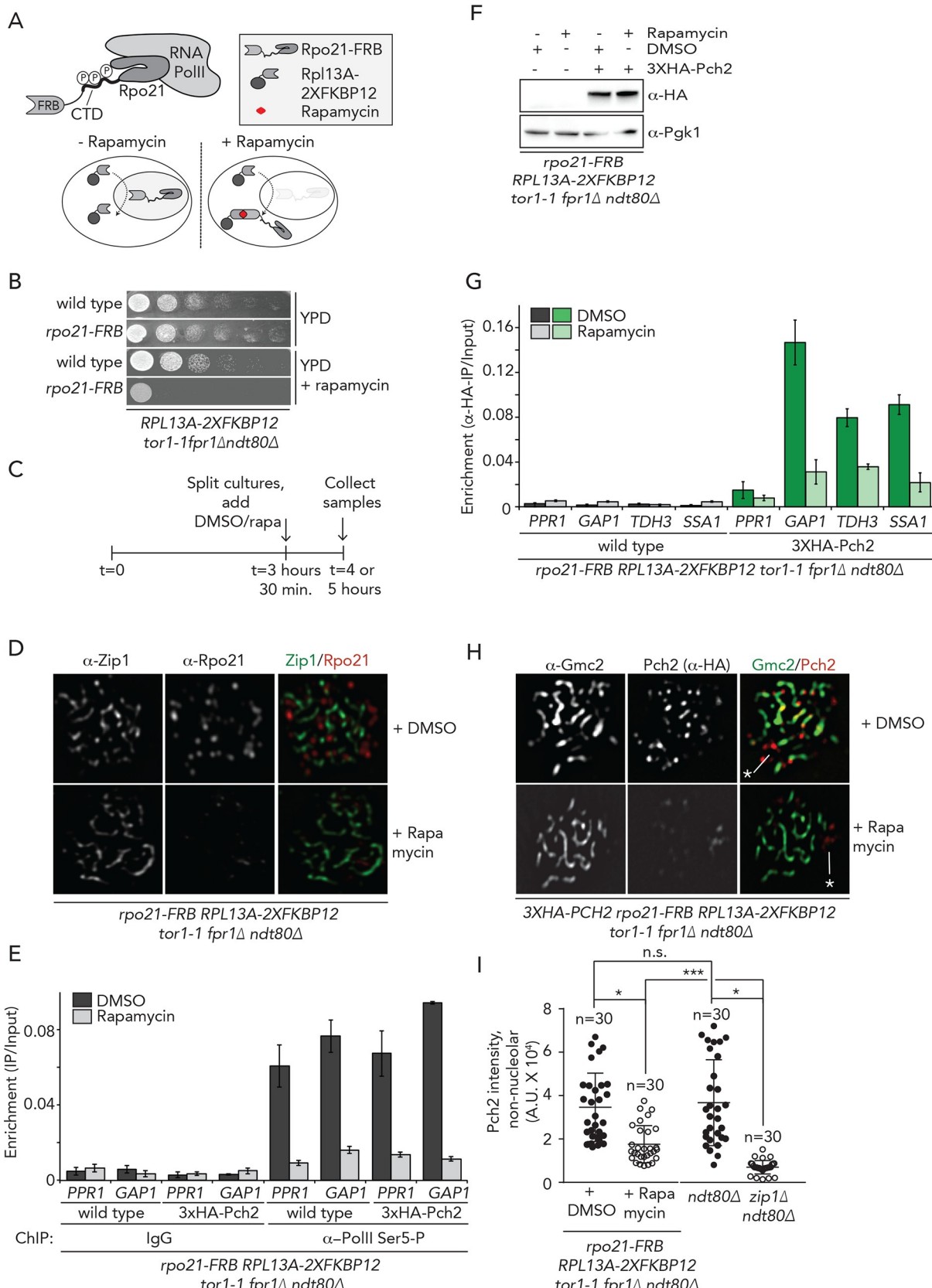

**Fig 3. Transcription is required for recruitment of Pch2.** A. Schematic of the anchor away system and Rpo21 within RNAPII. B. Dilution series of wild type and *rpo21-FRB* anchor away strains, grown on YPD or YPD + rapamycin. C. Schematic of treatment regimens used for anchor away experiments in D-I, and S5G Fig. D. Representative images of immunofluorescence of meiotic chromosome spreads in the *rpo21-FRB* anchor away cells treated with DMSO or rapamycin. Chromosome synapsis was assessed by α-Zip1 staining. E. ChIP-qPCR analysis of active transcription (α−Rpo21-phospho-Ser5) in *rpo21-FRB* anchor away cells treated with DMSO or rapamycin at *PPR1* (primer pairs: GV2390/GV2391) and *GAP1* (primer pairs: GV2597/GV2598). Error bars represent standard error of the mean of at least three biologically independent experiments performed in triplicate. F. Western blot analysis of 3XHA-Pch2 in *rpo21-FRB* anchor away cells treated with DMSO or rapamycin. G. ChIP-qPCR analysis of 3XHA-Pch2 in *rpo21-FRB* anchor away cells treated with DMSO or rapamycin at *PPR1* (negative control) (primer pair: GV2390/GV2391), *GAP1* (primer pair: GV2597/GV2598), *HOP1* (primer pair: GV2607/GV2608), *TDH3* (primer pair: GV2591/GV2592), and *SSA1* (primer pair: GV2587/GV2588). Error bars represent standard error of the mean of at least three biologically independent experiments performed in triplicate. H. Representative images of immunofluorescence of meiotic chromosome spreads in the 3XHA-Pch2 expressing *rpo21-FRB* anchor away cells, treated with DMSO or rapamycin. Chromosome synapsis was assessed by α-Gmc2 staining. White line and asterisk indicate the position of the nucleoli. I. Quantification of non-nucleolar Pch2 intensity per spread nucleus for the immunofluorescences shown in H, treated with DMSO or rapamycin and from the immunofluorescence shown in S5H Fig (*ndt80Δ* and *ndt80Δzip1Δ* cells collected at 4 hours after induction into the meiotic program.) * indicates a significance of p = 0.01–0.05, Mann-Whitney U test. See S3 Table for exact p-values.

antibodies against phosphorylated-Serine 5 of Rpo21 in meiotic cells following addition of rapamycin. This approach showed that the relative occupancy of RNAPII within *PPR1* and *GAP1* coding regions was substantially reduced after 30 minutes rapamycin treatment in cells expressing Rpo21-FRB (Fig 3E). Taken as a whole, these data demonstrate that RNAPII is depleted from the nucleus under this treatment regimen. Interestingly, ChIP analysis revealed that, whereas the protein levels of Pch2 were unaffected (Fig 3F), Pch2 association with *GAP1*, *TDH3* and *SSA1* was substantially reduced in Rpo21-FRB-tagged strains treated with rapamycin for 30 minutes (Fig 3G), confirming the fact that Pch2 association to defined chromosomal regions depended on RNAPII transcriptional activity.

We next addressed whether a reduction of Pch2 binding to these regions upon transcriptional inhibition could be corroborated through independent, cytological methods. For this, we performed immunofluorescence on spread chromosomes to quantify the chromosomal association of Pch2 during meiotic G2/prophase, and found that a brief inhibition (*i.e.* 30 minutes) of active transcription (via Rpo21-FRB nuclear depletion) triggered a significant reduction of Pch2 chromosome-associated foci within synapsed chromosome regions (as identified by staining with the SC component Gmc2 [46]) (Fig 3H and 3I). Treatment of cells with Rapamycin for a longer period (*i.e.* 90 minutes, S5A Fig) led to similarly reduced chromosomal levels of Pch2 (S5G Fig). The loss of Pch2 from synapsed chromosomes under both these conditions is less penetrant as compared to the loss of Pch2 from chromosomes that is seen in *zip1Δ* cells [18] (Fig 3I and S5H Fig). The nucleolar pool of Pch2 (identified by the typical nucleolar morphology and lack of association with SC structures) was not significantly affected by RNAPII inhibition (S5I Fig), suggesting that RNAPII-dependent transcription specifically promotes non-nucleolar recruitment of Pch2. This observation is in agreement with the association of Pch2 with RNAPI-dependent transcriptional activity within the rDNA, as suggested by our ChIP-seq analysis (see above). Together, these data identify active RNAPII-dependent transcription as a factor that positively contributes to the recruitment of Pch2 to euchromatic chromosome regions during meiotic G2/prophase.

We sought to better understand the recruitment of Pch2 to chromosomes, also in light of the connection with transcription. For this, we focused our attention on Orc1, a factor involved in Pch2 recruitment to the nucleolus [30, 39]. Orc1 is a component of the Origin Recognition Complex (ORC), a six subunit (Orc1-6) hexameric AAA+ ATPase (Fig 4A), and we recently showed, using *in vitro* biochemistry, that Pch2 directly interacts with ORC [47]. The first step in DNA replication occurs when ORC recognizes and directly binds to hundreds of origins of replication (also known as autonomously replicating sequences (ARSs)) across the genome (reviewed in [31]). Given the connection between ORC and Pch2 and the well-

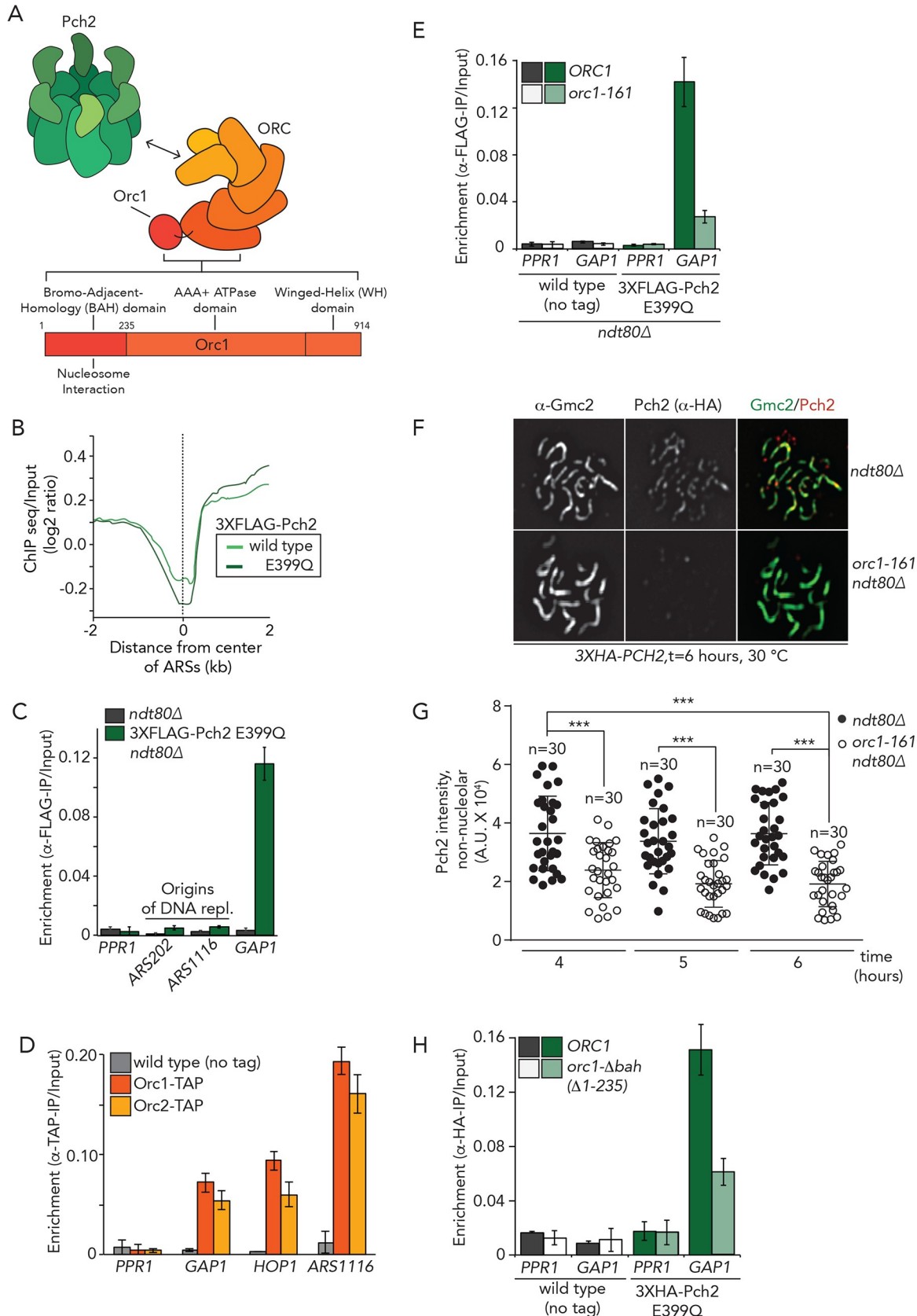

**Fig 4. Interplay between Pch2, Orc1 and transcription.** A. Schematic of Pch2 and ORC, including the domain organization of Orc1. B. 3XFLAG-Pch2 and 3XFLAG-Pch2-E399Q binding (ChIP-seq normalized against the input) plotted relative to the center of ARSs. C. ChIP-qPCR analysis of 3XFLAG-Pch2-E399Q at *PPR1* (primer pair: GV2390/GV2391), *GAP1* (primer pair: GV2597/GV2598), *ARS202* (primer pair: GV2583/GV2584) and *ARS1116* (primer pair: GV2577/GV2578) during meiotic G2/prophase (t = 4 hours). Error bars represent standard error of the mean of at least three biologically independent experiments performed in triplicate. D. ChIP-qPCR analysis of Orc1-TAP and Orc2-TAP at *PPR1* (primer pair: GV2390/GV2391), *GAP1* (primer pair: GV2597/GV2598), *HOP1* (primer pair: GV2605/GV2606), *ARS1116* (primer pair: GV2577/GV2578) during meiotic G2/prophase (t = 4 hours). Error bars represent standard error of the mean of at least three biologically independent experiments performed in triplicate. E. ChIP-qPCR analysis of 3XFLAG-Pch2-E399Q in *ORC1* or *orc1-161* background at *PPR1* (primer pair: GV2390/GV2391) and *GAP1* (primer pair: GV2597/GV2598) during meiotic G2/prophase (t = 4 hours). Experiments were performed at 30˚C. Error bars represent standard error of the mean of at least three biologically independent experiments performed in triplicate. F. Representative images of immunofluorescence of meiotic chromosome spreads in 3XHA-Pch2 expressing *ORC1* or *orc1-161* cells collected at 6 hours after induction into the meiotic program. Experiments were performed at 30˚C. Chromosome synapsis was assessed by α-Gmc2 staining. Experiments were performed at 30˚C. G. Quantification of non-nucleolar Pch2 intensity per spread nucleus of cells progressing synchronously through meiotic prophase at 4, 5 and hours (S6A and S6B Fig). *** indicates a significance of p≤ 0.001, Mann-Whitney U test. H. ChIP-qPCR analysis of 3XHA-Pch2-E399Q in *ORC1* or *orc1Δbah* background at *PPR1* (primer pair: GV2390/GV2391), *GAP1* (primer pair: GV2597/GV2598) during meiotic G2/prophase (4 hours). Error bars represent standard error of the mean of at least three biologically independent experiments performed in triplicate.

established association of ORC with origins, we first queried whether Pch2 was associated with origins. We plotted Pch2 occupancy ($\log_2$ genome-wide enrichment of Pch2 wild-type and E399Q) around the center of 626 predicted ARSs (from OriDB (http://cerevisiae.oridb.org); this list includes likely and confirmed ARSs). This analysis did not reveal significant enrichment for Pch2 at or around ARSs, and these regions seemed to rather display a local depletion of Pch2 binding (Fig 4B). The observation that Pch2 does not significantly associate with origins of replication was confirmed by ChIP-qPCR investigating individual origins (Fig 4C). We conclude that Pch2 is not detectably associated with origins of replication during meiotic G2/prophase, hinting that the interaction between Pch2 and Orc1 is established away from origins of replication [47].

Previous studies have described association of components of the replication machinery (including ORC) to actively transcribed, protein coding genes [48–50], and we thus considered the possibility that ORC exhibited a binding pattern similar to that of Pch2 during meiosis. The binding patterns of Pch2 showed some overlap with the reported association profile of ORC with coding genes in mitosis (*i.e.* 60 out of 436 peaks in Pch2 wild type and 4 out of 94 within the additional Pch2 E399Q binding sites) (S1 Table). We analyzed Orc1-TAP binding by ChIP-qPCR: as a positive control, we measured its association to *ARS1116*, and found Orc1 to be highly enriched at this site (Fig 4D), indicative of association of Orc1 with chromosomes during meiotic G2/prophase [39]. Strikingly, we also detected substantial binding of Orc1--TAP with *GAP1* and *HOP1*, but not *PPR1*, mirroring Pch2 binding behavior *(*Fig 4D and S6A Fig*)*. Similar results were seen for Orc2-TAP (Fig 4D). To explore the possibility that Orc1 functions upstream of Pch2 with respect to its localization – as has been suggested within the nucleolus/rDNA [30, 39] – we made use of a temperature-sensitive allele of *ORC1*, *orc1-161*. Pre-meiotic DNA replication is delayed for ~1 hour in *orc1-161* cells (S6B Fig), but meiotic progression appears otherwise normal [30]. This mutation severely diminished ORC association with origins of replication (as measured by ORC2-TAP ChIP; S6C Fig) (note that all *ORC1/ORC2* inhibition experiments were performed at 30˚C), likely due to a reduction in Orc1 protein levels (S6D Fig) [51]. The levels of Orc2-TAP at *GAP1* were reduced under these conditions (S6C Fig). We next performed Pch2 ChIP-qPCR in yeast strains harboring temperature-sensitive alleles of either *ORC1 (orc1-161)* or *ORC2 (orc2-1)*. As shown in Fig 4E, in *orc1-161* cells, Pch2 levels were strongly depleted at *GAP1*, suggesting that Orc1 contributes to the chromosomal association of Pch2, also outside the nucleolus. Pch2 protein levels appeared unaffected under these conditions (S6E Fig), and mRNA levels of selected Pch2-enriched sites

(*i.e. GAP1* and *HOP1*) were not changed in *orc1-161* cells (grown at 30˚C) (S6F Fig), arguing against indirect, transcriptional effects of Orc1 on Pch2 recruitment.

We next asked whether other subunits of ORC exhibited a similar connection to Pch2 recruitment. Orc2 was readily detected in *ORC2* cells but barely detected in *orc2-1* cells (S6G Fig). In contrast to the effects seen in *orc1-161*, ChIP-qPCR analysis of 3xHA-Pch2-E399Q revealed no effect of *orc2-1* on recruitment to *GAP1* (S6H and S6I Fig). Thus, although we cannot rule out that incomplete inactivation of Orc2 in *orc2-1* obscures effects on Pch2 recruitment, these data collectively suggest that Orc1 is involved in promoting the chromosomal association of Pch2.

Cytological analysis showed that Pch2 localization to the nucleolus/rDNA is severely impaired in *orc1-161* cells [30]. Our ChIP analysis indicated that Orc1 also contributes to recruitment at non-rDNA loci, and accordingly, using immunofluorescence on spread chromosomes we found that in *orc1-161* cells Pch2 localization was diminished within non-rDNA regions (Fig 4F and 4G). Due to the observed delay in premeiotic DNA replication of ~1 hour in *orc1-161* cells (see for example S6B and S7C Figs), a reduction in Pch2 association could be the result of indirect effects of delayed DNA replication on meiotic chromosome metabolism. We thus analyzed Pch2 chromosomal association at different times post induction of meiosis (*i.e.* after 4, 5 and 6 hours) and consistently found a significant reduction of Pch2 association with chromosomes in *orc1-161* cells (Fig 4F and 4G, and S7A and S7B Fig). Importantly, the significant reduction of Pch2 on meiotic chromosomes in *orc1-161* cells persisted, even when comparing *orc1-161* cells at t = 6 hours with *ORC1* cells at t = 4 hours. For these analyses, we focused on cells that appeared in the same G2/prophase stage, as judged by SC morphology (via Gmc2 staining), again hinting at a direct role for Orc1 in influencing efficient Pch2 chromosomal recruitment.

We cannot fully exclude that subtle defects in DNA replication upon *ORC1* inactivation, indirectly contribute to the observed effects on Pch2. However, together with the observation that Pch2 directly associates with ORC *in vitro* [47], these analyses indicate that ORC (and particularly Orc1) affects the localization of Pch2 at euchromatic chromosomal regions, likely in a direct manner. In addition, we provide evidence below that the ORC/Orc1 effect on Pch2 chromosomal association is correlated with its co-localization within regions of RNAPII-activity.

Orc1 contains a Bromo Adjacent Homology (BAH) domain, a nucleosome-binding domain [52] that contributes to ORC's ability to bind origins of replication [53] (Fig 4A). We and others have revealed a role for the BAH of Orc1 in controlling rDNA-associated functions [30, 54], and we interrogated whether occupancy of Pch2 to regions of transcriptional activity was affected in *orc1Δbah* cells. Indeed, deletion of the BAH domain of Orc1 reduced the association of Pch2 to sites of active transcription (Fig 4H and S7D and S7E Fig). Collectively, these results suggest that ORC/Orc1 may use its nucleosome-binding capacity (endowed by the BAH domain of Orc1) to bind to and recruit Pch2 to non-canonical (*i.e.* non-origin) genomic loci that are defined by transcriptional activity. In light of this, it is interesting to note that Orc1-BAH binding to nucleosomes – in contrast to a related BAH domain in Sir3 [55] – is insensitive to the acetylation state of Histone H4, which could allow effective engagement with (acetylated) nucleosomes within euchromatin [54].

Based on these observations, we explored a potential connection between Pch2 binding patterns and histone modifications. We generated a correlation matrix between Pch2 and published ChIP-seq datasets of a battery of chromatin marks [56](S8 Fig). Since comprehensive meiosis-specific maps of these marks are not available, we performed this analysis with maps generated from vegetatively growing cultures. This analysis revealed the strongest correlations between Pch2 and several modifications that are associated with active transcription [57], such

as Histone H3K4-dimethylation and Histone H3S10-phosphorylation. Interestingly, we also detected a correlation with Histone H4K20-monomethylation. Dimethylation of Histone H4K20 is a determining factor in driving ORC association with metazoan origins of replication through the direct recognition of this modification by the BAH domain of metazoan Orc1 [58]. Although budding yeast Orc1-BAH does not appear to recognize this modification [58], and Histone H4K20-dimethylation has not been detected in budding yeast, this finding might point to a possible connection between Pch2, Orc1, and Histone H4K20-monomethylation. In addition, Pch2 weakly correlated with Histone H3K79-monomethylation (but to a much lower extent with Histone H3K79-di, and tri-methylation). The Histone H3K79 methyltransferase Dot1 influences Pch2 chromosomal recruitment [22, 23], and the affinity of the BAH domain of Orc1 with nucleosomes is likely sensitive to modification of Histone H3K79 [55] [54]. The observed correlation between Pch2 binding and Histone H3K79-monomethylation thus potentially suggests that Dot1 activity influences Pch2 association to sites of active transcription.

In *pch2Δ* cells, abundance of Hop1 on synapsed chromosomes is increased [15, 18, 59] (S9A Fig). In addition, phosphorylation of Hop1 [60] persists in *pch2Δ* cells [15](S9B Fig). Likewise, conditions that impair Pch2 localization on chromosomes lead to increases in Hop1 chromosomal abundance and phosphorylation status [15, 18, 24]. We used cytological and ChIP-based approaches to test effects of acute removal of Pch2 from regions of transcriptional activity (via our Rpo21-FRB-based system) on Hop1 chromosomal abundance and function. First, we quantified Hop1 chromosomal association with synapsed chromosomes under conditions where RNAPII-inhibition caused diminished localization of Pch2 (*i.e.* 30 minutes long exposures to rapamycin) (Fig 5A and 5B, see also Fig 3H and 3I). Under these conditions, we did not observe significant changes in Hop1 chromosomal association. We exposed cells to rapamycin for longer periods (*i.e.* 90 instead of 30 minutes, see also S5G Fig), and also under these conditions, we did not observe alterations in the abundance of Hop1 on synapsed chromosomes (S9C Fig). These experiments were all performed in early meiotic G2/prophase (*i.e.* from 3:30 to 4 or 5 hours into the meiotic program). We also investigated effects later in meiotic G2/prophase (*i.e.* after 8 hours in *ndt80Δ* cells, to prevent exit from G2/prophase), when most cells contain fully synapsed SCs, and thus contain significant amounts of Pch2 on chromosomes. Under these conditions, inhibition of RNAPII equally did not lead to significant effects on Hop1 chromosomal recruitment (Fig 5B and S9D Fig). We next investigated Hop1 chromosomal levels by ChIP-qPCR, around a known Hop1 binding site in wild type and *pch2Δ* cells [44], but again did not find effects of acute inhibition of RNAPII on Hop1 chromosomal abundance at a selected locus (S9E and S9F Fig). We also probed the accumulation of the phosphorylated version of Hop1 (identified by a slower migrating band on SDS-PAGE gels), which reports on Hop1 'activation' on chromosomes [44]. In accordance with earlier observations, we did not observe differences in the amount of phosphorylated Hop1 under conditions of RNAPII depletion (Fig 5C and 5D and S9B Fig). Based on all cumulative experiments, we conclude that impairing the chromosomal association of Pch2 by inhibiting its recruitment to sites of active RNAPII-transcription does not lead to the detectable effects on the chromosomal association and function of Hop1. We also queried whether *ORC1* inactivation (which triggers reduction of Pch2 from chromosomes to a similar level as RNAPII nuclear depletion (Figs 3H, 3I, 4F and 4G)), was associated with effects on Hop1 chromosomal accumulation. Similar to what we observed after inhibition of RNAPII, we found that in *orc1-161* cells (queried at t = 4 hours post induction), Hop1 chromosomal levels were not different to those seen in *ORC1* cells (S9G and S9H Fig). Importantly, under these conditions we did detect an increased localization of Hop1 within nucleolar DNA, as expected from the confirmed role of Orc1 in driving Pch2 localization to the nucleolus [30] [39].

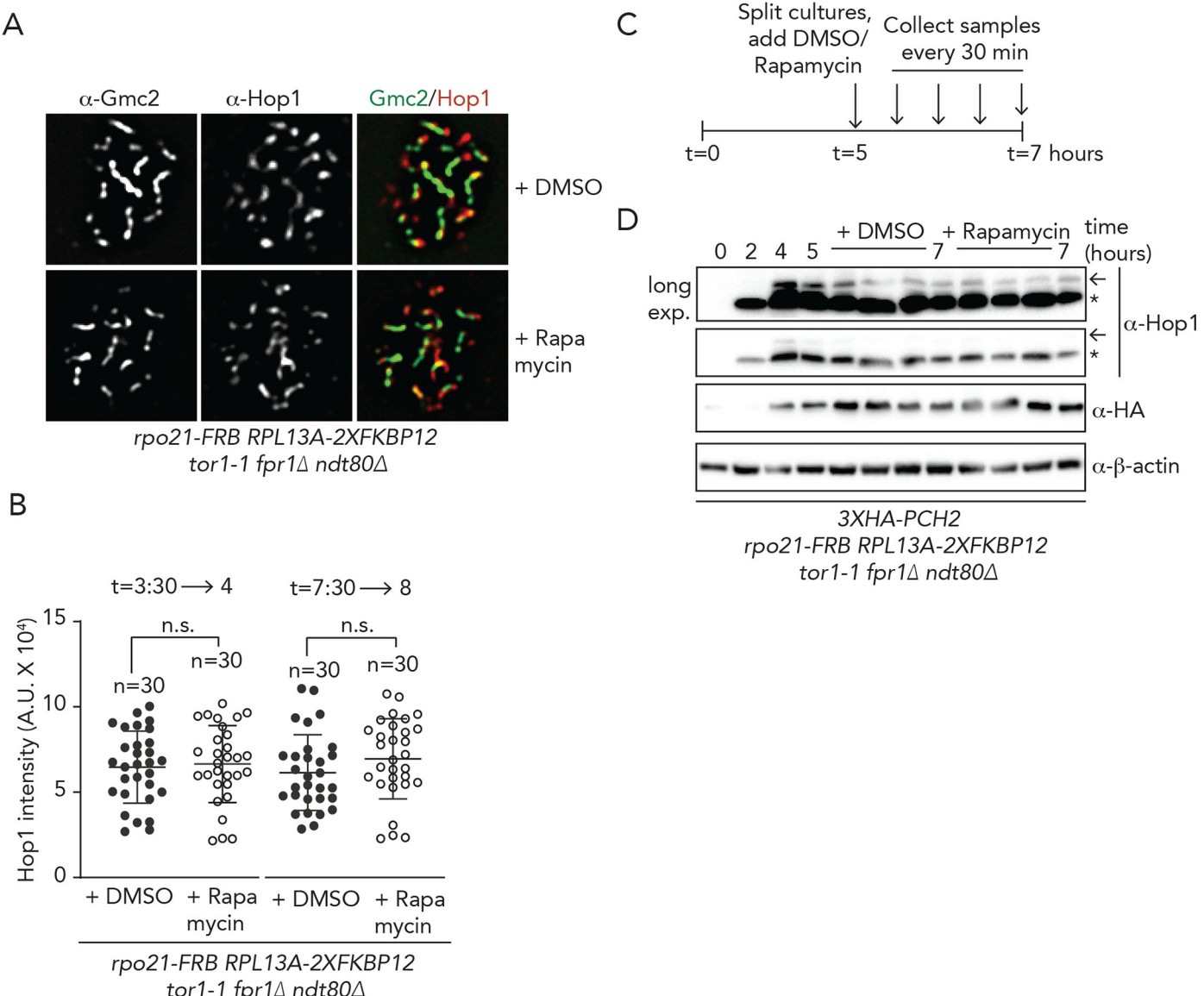

**Fig 5. Functional analysis of the transcriptional-associated Pch2 chromosomal population.** A. Representative images of immunofluorescence of Hop1 on meiotic chromosome spreads in *rpo21-FRB* anchor away cells, treated with DMSO or rapamycin, using the regimen indicated in Fig 3C. Chromosome synapsis was assessed by α-Gmc2 staining. B. Quantification of total Hop1 intensity per spread nucleus for the experiment shown in A and S9D Fig. n.s. (non-significant) indicates p>0.05, Mann-Whitney U test C. Schematic of treatment regimens used for anchor away experiment, as shown in D. D. Western blot analysis of Hop1 and Pch2 (α-HA) in *rpo21-FRB* anchor away *ndt80Δ* cells synchronously progressing through meiotic prophase, treated with DMSO or rapamycin, as indicated above the western blot image. Upon DMSO or rapamycin treatment samples were taken every 30 minutes. Arrow indicates phosphorylated Hop1, * indicates non-phosphorylated Hop1. See quantification of three independent western blot experiments on S4 Table.

Finally, we aimed to address further requirements for Pch2 association with selected regions of active RNAPII-dependent transcription. Most of the identified binding regions fall within genes that are also transcriptionally active in vegetatively growing cells (with the exception of a subset of meiosis-specific genes). In addition, Orc1/ORC is equally present in vegetatively growing cells. Therefore, we investigated whether ectopic expression of Pch2 – normally only expressed in meiosis – was sufficient to promote association with selected regions of binding, as identified in our ChIP analysis. We generated a galactose-inducible allele of Pch2

(*pGAL10-3HA-pch2-E399Q*) to induce Pch2-E399Q to protein levels comparable to those observed in meiotic G2/prophase (Fig 6A and 6B). Of note, in this allele, we removed the intron that is present in the *PCH2* to ensure proper mitotic expression and translation. ChIP-qPCR analysis revealed that, in contrast to the situation in meiotic cells, Pch2-E399Q was unable to associate with *GAP1*, *TDH3* or *DAL4* in mitosis (Fig 6C and S10A Fig). We confirmed that *GAP1* was actively transcribed during vegetative growth (Fig 6D). These results show that active transcription and presence of Orc1/ORC are not sufficient for the association of Pch2 with selected regions of active transcription, and instead suggest the presence of meiosis-specific factors that 'license' Pch2 binding. In agreement with such a model, we found that during meiosis, Zip1 was required for the recruitment of Pch2 to *GAP1* (Fig 6E and S10C and S10D Fig) mirroring the effect seen using cytological approaches [18] (Fig 3I and S5H Fig). As such, our results reveal a connection between transcription, ORC/Orc1 function and meiosis-specific chromosome organization that allows the recruitment of a specific chromosomal pool of Pch2 (Fig 6F).

## Discussion

The AAA+ protein Pch2 controls meiotic DSB formation, influences crossover recombination, mediates a meiotic G2/prophase checkpoint and is involved in chromosome reorganization upon chromosome synapsis (reviewed in [20]). For many functions, the chromosomal association of Pch2 has been postulated to be crucial. During meiotic G2/prophase Pch2 is enriched within the nucleolus/rDNA, and is also detected on synapsed chromosomes [18]. Chromosome synapsis is mediated by the dynamic polymerization of the synaptonemal complex (SC), whose formation is mostly nucleated at sites of crossover recombination [10–12]. Synapsis-dependent recruitment of Pch2 is abolished in cells that lack Zip1, the central element of the SC [13, 14, 18]. In addition to Zip1, other factors influence the chromosomal distribution of Pch2: Sir2 and Orc1 promote the nucleolar localization of Pch2 [22, 30], whereas Dot1 influences global chromosomal abundance of Pch2 [22, 23].

Here, we present a comprehensive analysis of the chromosomal association of Pch2 via genome-wide ChIP-seq. We reveal that Pch2 is associated with a subset of actively transcribed RNAPII-dependent genes. We perform several experiments and analyses to ascertain that these binding patterns are biologically meaningful and not caused by ChIP-associated artefacts [36] (see also Supplementary Data). Recruitment of Pch2 is dependent on active RNAPII-dependent transcription, but transcriptional strength *per se* is likely not the sole determining factor. For example, many actively transcribed genes do not recruit significant amounts of Pch2. If not solely determined by transcriptional strength and specific DNA content, what are additional factors that influence Pch2 association? We found that Orc1, a component of ORC, is required for Pch2 association with actively transcribed genes. Our data further indicate that the requirement of Orc1 to recruit Pch2 to sites of active transcription is likely independent of an effect of Orc1 on the transcriptional activity of these specific genes. In the future, it will be interesting to gain more insight into the interplay and function dependencies between active RNAPII-transcription, Orc1 recruitment, and Pch2 association.

The connection between Pch2 and Orc1 at genomic regions that are distinct from origins of replication further underscores the non-canonical role played by Orc1 during meiotic G2/prophase [47]. In addition, since Orc1 is also involved in nucleolar recruitment of Pch2 [30, 39], these findings hint at a common biochemical foundation that underlies recruitment of Pch2 to diverse chromatin environments. A recent study did not detect a role for Orc1 on the chromosomal (non-nucleolar) recruitment of Pch2 [39], and differences in the experimental approaches that were used to interfere with Orc1 function might underlie this discrepancy.

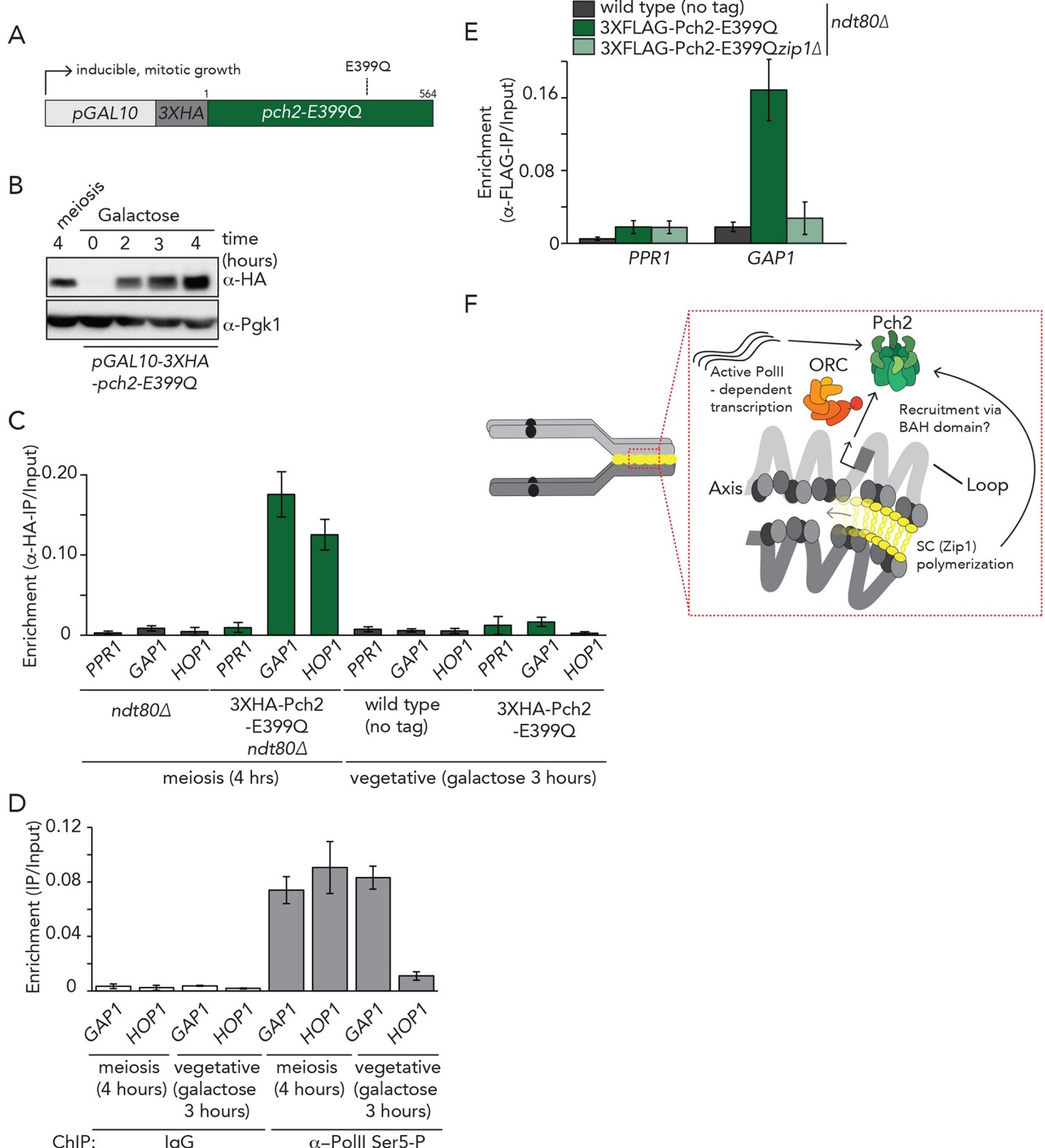

**Fig 6. Requirements of Pch2 binding in mitosis and meiosis.** A. Schematic of allele used for galactose-inducible mitotic expression of 3XHA-Pch2-E399Q. B. Western blot analysis of meiotic (*i.e.* endogenous) expression during meiotic G2/prophase at 4 hours after induction into the meiotic program and ectopic expression of *pGAL10-3XHA-pch2-E399Q*. Hours of treatment with galactose are indicated. C. ChIP-qPCR analysis of 3XHA-Pch2-E399Q at *PPR1* (primer pair: GV2390/GV2391), *GAP1*

(primer pair: GV2597/GV2598) and *HOP1* (primer pair: GV2605/GV2606) during in meiosis and mitosis. Time (hours) is indicated. Error bars represent standard error of the mean of at least three biologically independent experiments performed in triplicate. D. ChIP-qPCR analysis of active transcription (α–Rpo21-phospho-Ser5) *GAP1* (primer pair: GV2597/GV2598), *HOP1* (primer pair: GV2605/GV2606) in meiosis and mitosis. Hours are indicated. Error bars represent standard error of the mean of at least three biologically independent experiments performed in triplicate. E. ChIP-qPCR analysis of 3XFLAG-Pch2-E399Q at *PPR1* (primer pair: GV2390/GV2391) and *GAP1* (primer pair: GV2597/GV2598) in wild type or *zip1Δ* cells during meiotic G2/prophase (t = 4 hours). Error bars represent standard error of the mean of at least three biologically independent experiments performed in triplicate. F. Model depicting the interplay between Pch2 binding, active transcription, Orc1 and chromosome organization.

Specifically, we used a temperature-sensitive allele of *ORC1* (*orc1-161*) to deplete Orc1, whereas Herruzo *et al.* [39] employed an auxin degron-based method, and conceivably, depletion timing and efficiencies of Orc1 could differ under these conditions. Alternatively, this difference could be explained by the use of distinct strain backgrounds (SK1 versus BR): the chromosomal (*i.e.* non-nucleolar) localization of Pch2 in BR strains has been noted to be less pronounced as compared to Pch2 chromosomal localization observed in SK1 strains, used here (for example [61] [39]). In addition, BR and SK1 strains show quantitative differences in Pch2-dependent phenotypes (such as the ability to arrest in response to chromosome synapsis defects observed in *zip1Δ* cells, *e.g.* see [18] [32]). We speculate that these observations could reflect strain-specific differential reliance on chromatin recruitment and functionality of Pch2 on certain chromosomal recruitment pathways.

We show here that the Bromo-Adjacent Homology BAH-domain of Orc1, a nucleosome-binding module contributes to targeting of Pch2. BAH domains are readers of chromatin state [52]. A structural characteristic of a related BAH domain (*i.e.* that of budding yeast Sir3) is that nucleosome association is sensitive to Dot1-mediated Histone H3 K79 methylation (H3K79me) [55]. Interestingly, Dot1 activity (and H3K79 methylation state) is important for Pch2 localization along chromosomes [22, 23]. Dot1 activity is associated with active RNAPII transcriptional activity (reviewed in [62]), and RNAPII transcription-associated Dot1 activity may thus affect binding patterns of Pch2, potentially through Orc1 BAH domain-mediated nucleosome interactions. Of note, by analyzing correlations between our Pch2 data sets with genome-wide maps of several histone modifications [56], we found a relatively higher correlation between genome-wide Pch2 binding and H3K79-monomethylation patterns. We thus speculate that epigenetic status (*i.e.* specific chromatin modifications, such as Dot1-dependent H3K79-monomethylation) could contribute to Pch2 recruitment within euchromatin. However, we note that the correlation between Pch2 binding and Dot1-dependent modifications was relatively weak. Dot1 activity limits the general association of Pch2 with chromatin (as judged by cytological assays) [22, 23] and it thus remains unclear whether a contribution of Dot1 to transcription-associated Pch2 is a consequence of direct biochemical dependencies. Future biochemical and genome-wide association analysis should be able to provide more insight into the connection between nucleosomes, H3K79-methylation, and Pch2-ORC/Orc1.

Previous work has established that Pch2 localizes to the nucleolus [22], the site of RNAPI-driven rDNA transcription. We note that the nucleolus is a membrane-less, self-organized nuclear compartment that exhibits properties of a liquid-liquid phase separated nuclear condensate (reviewed in [63]). Interestingly, recent work revealed the existence of RNAPII-specific nuclear condensates that influence transcriptional regulation (reviewed in [37]). In the future, it will be interesting to investigate whether the biochemical properties of transcriptional condensates relate to (shared) characteristics for the recruitment and function of Pch2 at RNAPI- and RNAPII-transcriptional hubs.

Meiotic chromosomes are organized into a typical loop-axis structure shaped by active transcription [3, 4, 9, 64–69]. Our genome-wide mapping revealed that the Pch2 binding patterns are strikingly distinct from the stereotypical binding patterns of meiotic axis factors (*i.e.*

Hop1, Red1 and Rec8) [8, 9]. We suggest that the transcription-associated pool of Pch2 is not directly associated with chromosome axis sites and is instead associated with genes in loop regions. This is somewhat surprising since *i*) the only identified client of Pch2, Hop1 is a component of the axis, and *ii*) assembly of the SC, a structure that is assembled directly on the axis factors, is important to allow Pch2 recruitment to chromosomes. It is conceivable that the interaction between Pch2-associated sites and the chromosome axis is highly dynamic and flexible, and revealing the chromosomal behavior of Pch2 in living cells will be important to understand the organizational principles of Pch2 recruitment onto meiotic chromosomes.

We found clear evidence that both active transcription and Orc1 function promote the recruitment of Pch2 to meiotic chromosomes. Surprisingly, we found that impairing this recruitment pathway was not associated with detectable effects on Hop1 chromosomal association and function. We cannot currently rule out that incomplete inhibition of Pch2 association under the used conditions precludes us from exposing an effect on Hop1 function. Indeed, the effect of RNAPII/Orc1 inactivation on Pch2 recruitment was partial when compared to the (apparently complete) effect seen in *zip1Δ* [18] (see Figs 3I and 4G). Nonetheless, our results could indicate that merely disrupting recruitment of Pch2 to meiotic chromosomes is not sufficient to trigger defects in the dynamic association of Hop1 with chromosomes. Recent work, using an Orc1-degradable allele [39], also did not detect an effect of Orc1 inactivation on Hop1 levels and function.

Additional processes, such as chromosome restructuring and/or post-translational modifications of axis factors (*i.e.* phosphorylation of *ASY1*, the *Arabidopsis thaliana* homolog of Hop1) [24, 29] influence the functional interplay between Pch2 and Hop1 on chromosomes. Under conditions where effects are seen on Hop1 chromosomal association (and Pch2 recruitment) (*i.e. zip1Δ*, *zip1-4LA*, and *top2-1* [15, 18, 21, 24]*)* such additional requirements would be fulfilled, potentially in contrast to the conditions we describe here.

Alternatively, the pool of Pch2 identified here could play a role that is distinct from the canonical role of Pch2 in the removal of the bulk of Hop1 from meiotic axis sites. Accordingly, interference with recruitment of this Pch2 population (via transcriptional inhibition or Orc1 inactivation) would not be expected to affect the chromosomal association of Hop1. When considering the localization patterns of Pch2 described here, it is thus possible to speculate that more than one chromosomal population of Pch2 exists, of which one is recruited to transcriptionally active regions. In that scenario, a population of Pch2 that is directed to chromosome axis sites (that can possibly not be detected using ChIP-based approaches) might be responsible for the majority of Hop1 removal upon chromosome synapsis. However, under such a scenario, one would expect to be able to detect quantitative effects on Pch2 chromosome foci (as judged by immunofluorescence), where individual foci would be differentially affected by Rpo21/Orc1 inactivation. We, however, did not observe such differential behavior, and instead observed a general decrease in intensities of chromosomal Pch2 foci upon inactivation of Rpo21/Orc1 (for example, see Figs 3H and 4F).

We considered a role for Pch2 in transcriptional regulation, and tested this by comparing RNA-seq datasets from *pch2Δ ndt80Δ* and *ndt80Δ* cells. However, we found that these datasets were highly similar, especially when focusing on Pch2-associated genes or meiosis-specific genes (S11A and S11B Fig and S5 Table), suggesting that Pch2 does not affect transcription of genes that are directly associated with. For example, although Pch2 is associated with the *HOP1* gene, we found no significant difference between *HOP1* mRNA levels in *ndt80Δpch2Δ* relative to *ndt80Δ* cells by differential expression analysis (log$_2$ fold: 0,190, *p*-value = 0.46, see S5 Table).

What is the role of the transcription-associated pool of Pch2? The connection between Pch2 and synaptonemal complex formation and the contribution of Zip1 to the recruitment of

the transcription-associated Pch2 population (Fig 6) might provide a clue. SC polymerization along synapsing chromosomes has been proposed to trigger mechanical reorganization [70] [24], which might influence large-scale topological organization of loop-axis structures, and Pch2 localization. Pch2 also influences chromosome organization upon synapsis [15]. Understanding how Pch2 is recruited to transcriptionally active regions, and how this is mechanistically linked to SC function and Orc1 should provide clues to the roles of the chromatin-associated pool of Pch2 we describe here. It will be interesting to understand the link between transcription, chromosome topology, organization and Pch2 recruitment. In addition, Pch2 impacts global DSB activity [44], and the specific recruitment of Pch2 to defined chromosomal regions might conceivably impact local DSB patterning and potentially, genome stability (within actively transcribed regions). Finally, the meiotic functions of Pch2 have been attributed to its biochemical effects on Hop1 (reviewed in [20]), but it is conceivable that the AAA + ATPase activity of Pch2 acts on additional (potentially HORMA domain-containing) clients whose roles might be related to transcription-associated processes. Clearly, biochemical and genome-wide analysis, coupled to functional investigation will be needed to reveal Pch2 function and regulation during meiotic G2/prophase.

In conclusion, we have used genome-wide methodology to reveal a hitherto unknown relationship between Pch2, active transcription and Orc1, which influences the chromosome synapsis-driven recruitment of Pch2 to euchromatin during meiotic G2/prophase. Future work is needed to understand how the dynamic chromosome recruitment of this important AAA + ATPase contributes to meiotic DSB formation, recombination and checkpoint signaling.

## Materials and methods

### Yeast strains and growth conditions

All yeast strains used in this study were of the SK1 strain background, except for the strains harboring the galactose-inducible promoter (*pGAL10*) system, which are of the W303 background. The genotypes of these strains are listed in Supplementary Data. For experiments using mitotically cycling cells (as shown in Fig 5A–5D), cells were grown to saturation in YP-D/R medium ((1% (w/v) yeast extract, 2%(w/v) peptone, 0,1%(w/v) dextrose and 2%(w/v) raffinose)) at 30˚C. Induction of synchronous meiosis was performed as described in [30]. Cultures were diluted to an optical density at 600nm (OD600) of 0.4, grown for an additional 4 hours after which 2% galactose was added. Unless stated otherwise, samples of cells undergoing synchronous meiosis were collected 4 hours after incubation in sporulation (SPO) medium. Synchronous entry of cultures into the meiotic program was confirmed by flow cytometry-based DNA content analysis (see below). For experiments using temperature-sensitive strains, meiotic induction was performed as described in [30], except that cells were grown for up to 24 hours in pre-sporulation medium (BYTA) at the permissive temperature (23˚C). Meiotic cultures were then shifted to 30˚C. For the inhibition of global transcription (S5 Fig) 1,10- Phenanthroline (100 μg/mL in 20% ethanol, Sigma-Aldrich) [41] was added to cultures 3 hours after induction into the meiotic program. Cells were subsequently grown for one hour and harvested. For mitotic expression of Pch2-E399Q, the coding sequence of *pch2*-E399Q (lacking its intron) was cloned in a *URA3* integrative plasmid containing *pGAL10-3XHA*. The plasmid was integrated at the *URA3* locus. For expression of 3XFLAG-dCas9 in meiosis, *3XFLAG-dCas9-tCYC1* was cloned in a TRP1 integrative plasmid containing *pHOP1* to create *pHOP1-3XFLAG-dCas9-tCYC1*. The plasmid containing *3XFLAG-dCas9/pTEF1p-tCYC1* was a gift from Hodaka Fujii and obtained via Addgene.org (Addgene plasmid #62190) [71].

## Nuclear depletion via the anchor-away method

Rpo21 was functionally depleted from the nucleus using the anchor away technique [42, 45]. Briefly, Rpo21 was tagged with FKBP12-rapamycin-binding (FRB), and this allele was introduced in strains harboring the anchor away background (*RPL13A-2xFKBP12 fpr1Δ tor1-1*) [42]. Nuclear depletion of Rpo21-FRB was achieved by addition of rapamycin at a concentration of 1 μM. Exact treatment regimens are indicated per experiment (see figure legends). For viability assays, serial dilutions of mitotically growing yeast cells were spotted on solid YPD containing 1 μM rapamycin for 2 days.

## Co-immunoprecipitation and western blots

100mL of SPO cultures (OD600 1.9), were harvested at 3000 rpm for 3 min at 4˚C and washed once with ice-cold Tris-buffered saline (TBS) buffer (25 mM Tris–HCl, pH 7.4, 137 mM NaCl, 2.7 mM KCl). Cells were snap-frozen in liquid nitrogen and stored at -80˚C until further use. Cells were resuspended in lysis buffer (50mM Tris-HCl pH 7.4, 150mM NaCl, 1% Triton X-100, and 1mM EDTA) containing protease inhibitors and broken with glass beads using bead beater (FastPrep-24, MP Biomedicals; 2 X 60 seconds at speed 6.0, incubated on ice in between for 5 min). Chromatin was sheared by sonication using a Bioruptor (Diagenode), 25 cycles of 30 seconds on/off, high power at 4˚C. Lysates were clarified by centrifugation for 15 min at 16,000 x g at 4˚C. Lysates were then immunoprecipitated with α-TAP antibodies using magnetic beads (Invitrogen), washed 4 times with buffer containing detergent and another time with the same buffer without detergent. Beads were eluted in 1X loading buffer (50 mM Tris-HCl pH 6.8, 2% SDS, 10% glycerol, 1% β-mercaptoethanol, 12.5 mM EDTA, 0.02% bromophenol blue) and the supernatant resolved by SDS-PAGE followed by Western blotting. Protein extracts were prepared by using trichloroacetic acid (TCA) extraction protocol as previously described [30]. Samples were resolved by SDS-PAGE, transferred to nitrocellulose membranes and probed with the following primary antibodies diluted in 5% (w/v) nonfat-milk in TBS buffer + 0.1% Tween 20: α-Flag (Sigma-Aldrich, F3165, 1:1000), α-ORC2 (Abcam, 31930, 1:500), α-HA (BioLegend, 901502, 1:1000), α-TAP (Thermo Scientific, CAB1001, 1:2000), α-Pgk1 (Invitrogen, 459250, 1:5000), α-Rpo21 (BioLegend, 664906, 1:500), α-phosphoserine 5 Rpo21 (Thermo Scientific, MA518089, 1:1000), α-Histone-H3 (Abcam, AB1791, 1:1000), α-FRB (Enzo, ALX215-065-1, 1:500), α-Zip1 (Santa Cruz Biotechnology, YC-19, 1:1000), α-Hop1 (kind gift of Nancy Hollingsworth, Stony Brook University, Stony Brook, USA, 1:5000), α-β-Actin (Abcam, AB170325, 1:5000), α-Histone H2A (Active Motif, AB2687477, 1:1000) or α-ORC (kind gift of Stephen Bell, MIT, Cambridge, USA, 1:2000). Membranes were incubated with horseradish peroxidase-conjugated goat anti-rabbit IgG, anti-mouse IgG and donkey anti-goat IgG (Santa Cruz Biotechnology). Proteins were detected with ECL (GE Healthcare) using a digital imaging system Image-Lab (Bio-Rad).

## Chromatin Immunoprecipitation (ChIP)

For ChIP experiments, 100 mL SPO-cultures (OD600 1.9) were harvested 4 hours after entering meiosis, unless stated otherwise. Meiotic cultures or exponentially growing mitotic cultures were crosslinked with 1% methanol-free formaldehyde for 15 minutes at room temperature and the reaction quenched with 125 mM Glycine. Cells were washed with ice-cold TBS, snap-frozen and stored at –80˚C. Cells were resuspended and broken with glass beads using a bead beater, as described above. Chromatin was sheared using either a Branson Sonifier 450 (microtip, power setting 2, 100% duty cycle, 3X for 15sec, 2 min on ice in between) or using a Diagenode Bioruptor UCD 200 (25 cycles of 30 seconds on/off, high power at 4˚C). Cells were centrifuged at 13000 rpm for 10 min at 4˚C. 10% of sample was

removed for input. 550 µl of cell lysates were pre-incubated with the following antibodies for 3 hours at 4°C prior to overnight incubation under rotation with magnetic Dynabeads-protein-G (Invitrogen): for ORC ChIP, 1 µl of α-ORC and 1 µl of isotype control antibody (α-rabbit IgG (Bethyl, P120-101)). For TAP ChIP, 1 µl of α-TAP, and for HA ChIP, 1 µl of anti-HA. For RNA Pol-II ChIP, 1 µl of α-Rp021 or α-phosphoserine 5 Rpo21. Immunoprecipitates were incubated and washed as described above. For FLAG ChIP cells lysates were incubated with 30µl of 50% α-Flag-M2 affinity gel (Sigma-Aldrich, A2220) for 3 hours. Bound proteins were eluted using a 3XFLAG peptide (Sigma-Aldrich, F4799) as described in [72]. For Hop1 ChIP, 50mL SPO-cultures (OD600 1.9) were processed as above and cells were resuspended in 400µl FA-lysis buffer (50 mM HEPES-KOH, pH 7.5; 140 mM NaCl, 1mM EDTA, 10% Glycerol, 1% Triton X-100, 0.1% sodium deoxycholate, + protease and phosphatase inhibitors). Cell breakage and chromatin shearing were performed as above. Dynabeads-Protein-G (Invitrogen) were previously blocked overnight with a solution containing 1µg/µL BSA (Sigma) and 1µg/µL Yeast tRNA (Invitrogen) in TBS. Chromatin was incubated with 1 µl of α-Hop1 or 1 µl of the isotype control (rabbit IgG) for 3 h at 4°C prior to overnight incubation under rotation with blocked magnetic beads. Beads were washed as above with ice-col FA-lysis buffer. Subsequent steps (*i.e.* reversal of crosslinking, Proteinase-K and RNase-A treatments and final purifications and elutions) were performed as previously described in [73].

## ChIP-Seq library preparation

Preparation of paired-end sequencing libraries was performed using the Illumina TruSeq ChIP library preparation kit–Set-A (15034288), according to the manufacturer's guidelines. Ligation products were size-selected (250–300 bp) and purified from a 2% low-melting agarose gel using the MinElute Gel Extraction Kit (Qiagen). Ampure XP beads (Agilent) were used for cleanup steps and size selection. The final purified product was quantitated using Picogreen in a QuantiFluor dsDNA System (Promega). Paired-end sequencing (2 X 150bp) was performed on the Illumina HiSeq 3000 platform at the Max Planck Genome Centre (Cologne, Germany). The ChIP-seq raw data employed in this study are deposited at the NCBI Gene Expression Omnibus (http://www.ncbi.nlm.nih.gov/geo/), under accession no. GSE138429.

## Processing of ChIP seq data

Preliminary quality control of raw reads was performed with FastQC (http://www.bioinformatics.babraham.ac.uk/projects/fastqc/). Illumina raw reads were filtered for removal of low quality and duplicated reads, adapters and low-quality bases using the SAM tools (version 1.2). Paired-sequencing reads from three biological independent replicas (ChIPed DNA and their respective inputs from 3XFLAG-Pch2 and 3XFLAG-tagged-Pch2-E399Q expressing cells) were mapped to the yeast genome S288C (SacCer3) using *Bowtie2* (version 2.3.4.3) with default parameters. Non-unique reads were removed using SAMtools (Version 1.2). The dataset was pooled and the Model-based Alignment of ChIP-Seq (MACS2) program was used to call peaks of Pch2 occupancy using the input as a control ($P$-value = $e10^{-15}$). The data visualized on IGV [74] were normalized to the number of Reads per Kilobase per Million (RPKM) of total mapped reads. Bin size was set to 25. For analysis shown in S3A Fig, peak calling was performed as above for NLS-GFP ChIP-seq data (SRR2029413) [36]. For ChIP-seq analyses shown in Fig 2A and 2B, and S4A and S4B Fig, processed ChIP-seq data were retrieved from the Gene Expression Omnibus (GEO), access number: GSE69232 (Hop1: GSM1695721; Red1: GSM1695718; and Rec8: GSM1695724) [9]. For S8E Fig, processed ChIP-seq data were obtained from [44], GEO, access number: 105111,

(wild type; GSM2818425 and, *pch2Δ*; GSM2818432). Peaks were visualized on IGV [74]. For analysis shown in S2A Fig, datasets were from [36], retrieved from the National Center for Biotechnology Information (NCBI) Short Read Archive under accession number SRP030670. GFP_NLS ChIP-seq and inputs [36].

## RNA-Seq

10 mL of yeast undergoing meiosis at 4h (OD600 1.9) was harvested at 1000g for 5 min at 4°C. Cells were washed with ice-cold TE buffer and mechanically lysed with glass beads. Samples were centrifuged for 2 min at full speed and transferred to a new microcentrifuge tube. 1 mL of 70% ethanol was added to the homogenized lysate. Total RNA was then extracted and purified using the RNeasy RNA isolation kit (Qiagen), including treatment with DNase. RNA integrity was assessed by urea RNA-PAGE and quantitated with Quantifluor (Promega). Enrichment of mRNA, library preparation and sequencing was performed by the Max Planck-Genome Centre (Cologne, Germany). Briefly, rRNA was depleted using the Ribo-zero rRNA removal kit (Epicentre). mRNA was enriched using oligo-dT beads (New England Biolabs) and the cDNA library was prepared using the TrueSeq RNA sample preparation kit (Illumina) according to the manufacturer's instructions. Samples were then sequenced on an Illumina HiSeq-3000 instrument. The RNA-seq raw data employed in this study are deposited at the NCBI Gene Expression Omnibus (http://www.ncbi.nlm.nih.gov/geo/), under accession no. under accession no. GSE144835.

## RNA-Seq data analysis

Raw reads were firstly examined and quality trimmed using the FastQC package (http://www.bioinformatics.babraham.ac.uk/projects/fastqc/). Reads were filtered and mapped to the S288c genome assembly R64 (sacCer3) using Hisat2 (Version 2.1.0), with default parameter settings. Analyses and quantification of transcripts were performed using the HTSeq (Version 0.9.1) and DESeq2 (2.11.40.6), respectively. The number of mapped reads and comparison of transcripts expression level were estimated using TPM (Transcripts Per Million fragments) values. The $\log_2$ fold changes of transcripts (TPM) was used to estimate differential expression levels. Transcripts with $-\log_{10}$ of P-value $> 2$ and $\log_2$ fold changes $\geq$ or $\leq 2.0$ were considered as significantly differentially expressed. In S1E Fig, expression levels of *ndt80Δ* strains (normalized RNA-seq count data expressed as Transcription per Million—TPM) were stratified as low (0.5–10), medium (>10–1000) and high (>1000). For S11 Fig, transcripts presenting baseline expression below 5 (log2 = 2,32) were considered as non-expressed as defined by analyses of basal transcription levels of RNA-seq experiments performed in cells undergoing meiosis. For S2F Fig, the Pearson correlation was calculated with normalized read counts scores ($Log_{10}$) for each Pch2-wild-type binding gene. Expression values for each gene (TPM, $Log_{10}$) were obtained from the RNA-seq data (*ndt80Δ*), processed data available at GEO: GSE144835. For S11B Fig, the list of genes known to be involved in the budding yeast meiosis was retrieved from the KEGG pathway https://www.genome.jp/kegg/pathway.html [75].

## Computational analyses

To plot the enrichment within specific regions, annotation tracks, including the tracks from ChIP-seq data were converted into BED files using The UCSC Genome Table Browser (https://genome.ucsc.edu/cgi-bin/hgTables). Yeast tRNAs coordinates were obtained from SGD (http://www.yeastgenome.org). Autonomously Replicating Sequences (ARSs) (coordinates and sequences for the 626 (dubious ARSs were excluded from the analysis) origins were downloaded from the OriDB (version 2012 (http://www.oridb.org)). Distance map plots

analyses were performed with the bamCompare tool (Version 2.5.0) [76] with the following parameters: bin size 50, pseudo count 1.0, limiting the operation to specific regions (Pch2 binding sites, ARSs, tRNAs). The Pch2-ChIP-seq data and the inputs were normalized and compared to compute the $\log_2$ ratio of the normalized number of reads. The enrichment was calculated as $\log_2$ ratio of the normalized number of reads: All analyses were carried within the Galaxy platform [76]. Pairwise correlation (Spearman) among Pch2-ChIP seq and Input replicas was estimated using multiBamsummary tool (version 2.5.0). The bin size was set to 200bp and regions with very large counts (which artificially increases correlation) were removed. To compute the correlation between different ChIP-seq datasets, mapped and filtered reads (ChIP-seq data mapped reads normalized against their respective inputs), were computed using the parameters described above. Correlation matrices were created for Pch2-ChIP-seq and Hop1, Red1, Rec8 (GSE69232) [9], and for a different set of histone modifications obtained from [56] (from vegetatively growing yeast strains; GEO61888 and SRA accession numbers within for both ChIP and input data): H4K20me, H3K4me, H3S10phospho, H4K16, H36Kme2, H3K26me2, H2ak5ac, H4k5ac, H4K8ac, Htz1, H3K79me, H3K18ac, H3K4me3, H3K4me3, H3K27ac, H3K4me2, H3K56ac, H3K23ac, H3K14ac, H3K4ac, H2As129phospho, H4R3me2 H3K79me3, H3K79me2. The Spearman correlation matrix was generated with the plot correlation tool (version 3.3.0) [76]. Venn diagrams were generated using the web-based Venn diagram generator from http://jura.wi.mit.edu/bioc/tools/venn.php. To determine whether the Pch2 binding genes were enriched in certain functional categories, gene ontology analysis was conducted using the SGD GO term finder (molecular function) (https://www.yeastgenome.org/goTermFinder) at a p-value cut-off of 0.01.

## ChIP-qPCR

ChIP and Input samples were quantified by qPCR on a 7500 FAST Real-Time PCR machine (Applied Biosystems). The percentage of ChIP relative to input was calculated for the target loci as well as for the negative controls. Enrichment [relative to time untagged control or IgG (control)] was calculated using the $\Delta$Ct method as follows: $1/(2^{[Ct-Ctcontrol]})$. Primer sequences (including primer efficiency) covering the various loci are listed in the Supplementary Data.

## RNA extraction and RT-qPCR

For RNA extraction, 15 mL meiotic cultures were harvested and total RNA was extracted using the hot-acidic phenol method [77], with some minor modifications. Cells were resuspended in 600 µl of freshly prepared TES buffer (10 mM Tris-Cl, pH 7.5 10 mM EDTA 0.5% SDS). 600 µl of acidic-phenol (Ambion) was added and the solution was immediately vortexed vigorously for 30 seconds. Samples were incubated at 65°C for 90 min under rotation at 300 rpm. The solution was kept on ice for 10 minutes and spun down at 14000 rpm for 10 minutes at 4°C. The aqueous top layer was transferred to a new tube and 600 µl of chloroform was added and immediately vortexed. Cells were centrifuged as above after which the aqueous layer was transferred to a new pre-chilled eppendorf tube. RNA was precipitated overnight at -20°C with 2.5 volumes of 100% ethanol and 10% (v/v) sodium acetate, pH 5.4 and washed with 75% ethanol. After drying on ice, RNA was eluted with RNase free water and stored at -80°C. cDNA was generated using the superscript-III reverse transcriptase (Invitrogen) according to the manufacturer's protocol. Briefly, 1–2 µg of total RNA was used in a 20 µl reaction mixture using random primer mix or oligodT-20 (Invitrogen). Relative amounts of cDNAs of various genes were measured by real-time quantitative PCR (RT-qPCR) on a 7500 FAST Real Time PCR machine (Applied Biosystems). The experiment depicted on S6F Fig

was performed using the CFX-Connect Real Time PCR detection system (Bio-Rad). Expression of every gene was normalized to the expression of *18S* (RNA Pol I transcript) or β-Actin (*ACT1*) using the 2^-ΔΔCt method. Oligo sequences and their respective amplification efficiencies are available in Supplementary Data.

## Chromosome spreads and immunofluorescence

Chromosome spreading was performed as described in [15]. For immune staining, the following antibodies were used: α-HA (Roche, 11867423001, 1:200), α-Zip1 (Santa Cruz Biotechnology, YN-16, 1:500), α-Gmc2 (a kind gift of Amy MacQueen, Wesleyan University, Middletown, CT, USA, 1:200), α-Rp021 (BioLegend, 664906, 1:200) and α-Hop1 (homemade, 1:2000). α-Hop1 was raised against full-length 6-His-tagged Hop1 expressed in bacteria. Hop1 was purified via affinity purification followed by ion-exchange chromatography, and used for immunization. Antibody production was performed at the antibody facility of the Max Planck Institute of Molecular Cell Biology and Genetics (Dresden, Germany). DNA was stained with 4',6-Diamidine-2'-phenylindole dihydrochloride (DAPI). Images were obtained using a DeltaVision imaging system (GE Healthcare) using a sCMOS camera (PCO Edge 5.5) and 100x 1.42NA Plan Apo N UIS2 objective (Olympus). Deconvolved images (SoftWoRx software 6.1.l and/or z-projected) using the SoftWoRx software 6.1.l). were quantitated using Imaris Software (version 9.3.0) (Oxford Instruments). To estimate the intensity of Pch2 signal, we first created a surface by selecting the green channel (Gmc2) and excluded the foci (red channel–3XHA-Pch2) located outside of the selected surface. We then estimated the intensity solely covering the green surface. Estimation of intensity of the nucleolar Pch2 pool was performed by creating a region of interest surrounding the nucleolus and measuring the Pch2 signal not covering the Gmc2 surface. Hop1 intensity was carried out in a similar fashion with the quantification of Hop1 signal within the Gmc2 surface. All values were normalized against the background. Scatterplots were created using the Graphpad program (Prism) and statistical significance was assessed using a Mann-Whitney test.

## Supporting information

**S1 Text. Supplementary Data.**
(DOCX)

**S1 Fig.** A. Co-immunoprecipation experiment showing an interaction between 3XFLAG-Pch2 and Orc1-TAP. Pgk1 was used as a control. B. Spore viability analysis of the indicated strains. Cells were sporulated for 24 hours in liquid media. The total analyzed tetrads/strain are indicated. C. Representation of meiotic G2/prophase progression in used strain. SC appearance was detected using Gcm2. For each time point 125 cells were counted. D. Genome browser view representative images (RPKM; see also Material and Methods) of ChIP and input signals for 3XFLAG-Pch2 and 3XFLAG-Pch2-E399Q. Shown is a region of Chromosome *XII* (chromosomal coordinates (kb) are indicated) identical to ChIP-seq binding patterns as shown in Fig 1C. E. High resolution Genome browser view representative images (RPKM; see also Material and Methods) of 3XFLAG-Pch2 and Hop1 binding patterns (from [44]) across two selected chromosomal regions (chromosomes *XI* and *IX*), identical to ChIP-seq binding patterns as shown in Fig 1D. Chromosomal coordinates and gene organization are indicated.
(TIFF)

**S2 Fig.** A and B. Heatmap matrices depicting correlation analysis for Pch2 ChIP-seq and Input replicates as measured by the Spearman correlation coefficient (R-values within the

squares). C. Scatter plot correlation analysis of Pch2-ChIP-seq (Wild-type versus E399Q) normalized read counts (Log10—RPKM) measured by the Pearson correlation coefficient R-values (upper right corner). D. Normalized read counts (ChIP/input) for 3XFLAG-Pch2 and 3XFLAG-Pch2-E399Q during meiotic G2/prophase, as determined by ChIP-seq. E. Comparison between expression strength of Pch2-associated genes and the transcribed genes from our mRNA dataset (*ndt80Δ* cells) binned into high, medium and low expression strength (following previously established procedures [35]). F. Correlation between normalized reads (log$_{10}$) of 3XFLAG-Pch2 individual binding sites and mRNA expression values (Transcription Per Million). G. ChIP-qPCR analysis of 3XFLAG-Pch2-E399Q at *PPR1* (primer pair: GV2390/ GV2391) and *GAP1* (primer pair: GV2597/GV2598) during meiotic G2/prophase (4 hours). H. ChIP-qPCR analysis of active transcription (α-phosphoserine 5 Rpo21) at *PPR1* (primer pair: GV2390/GV2391) and *GAP1* (primer pair: GV2597/GV2598) during meiotic G2/prophase (t = 4 hours). I. Western blot analysis of expression of 3XFLAG-Pch2, 3XFLAG-Pch2-E399Q and 3XFLAG-*Δ*NTD-Pch2 during meiotic G2/prophase (t = 4 hours). J. ChIP-qPCR analysis of 3XFLAG-Pch2, 3XFLAG-Pch2-E399Q and 3XFLAG-*Δ*NTD-Pch2 at *PPR1* (primer pair: GV2390/GV2391), *GAP1* (primer pair: GV2597/GV2598), *HOP1* (primer pair: GV2607/GV2608), *TDH3* (primer pair: GV2591/GV2592), and *SSA1* (primer pair: GV2587/ GV2588) during meiotic G2/prophase (t = 4 hours). Error bars represent standard error of the mean of at least three biologically independent experiments performed in triplicate. (TIFF)

**S3 Fig.** A. Venn diagram comparing 3XFLAG-Pch2 binding peaks and HyperChIPpable regions as described by [1]. B. 3XFLAG-Pch2 and 3XFLAG-Pch2-E399Q ChIP-seq normalized read counts enrichment normalized to inputs (log$_2$). Datasets were aligned relative to centre of tRNAs. C. ChIP-qPCR analysis of 3XFLAG-Pch2 and 3XFLAG-dCas9 at *PPR1* (primer pair: GV2390/GV2391) and *GAP1* (primer pair: GV2597/GV2598) during meiotic G2/prophase (4 hours). Error bars represent standard error of the mean of at least three biologically independent experiments performed in triplicate. D. Western blot analysis of 3XFLAG-Pch2 and 3XFLAG-dCas9 as used in C. (TIFF)

**S4 Fig.** A and B. Representative images of ChIP-seq binding patterns for 3XFLAG-Pch2, Hop1, Red1 and Rec8. Data for Hop1, Red1 and Rec8 are from [9]. Shown is the entire chromosome *I* (A) and a region of chromosome *III (B)* (chromosomal coordinates (kb) are indicated). (TIFF)

**S5 Fig.** A. Schematic of 1,10- Phenanthroline treatment regimen as used for C-D B. mRNA quantification of *GAP1* (primer pair: GV2597/GV2598)/*18S (*primer pair: GV2717/ GV2718) in cells (wild type or 3XFLAG-Pch2-E399Q) treated with mock (20% EtOH) or 1,10- Phenanthroline (final concentration is 2% EtOH and 100 μg/ml 1,10- Phenanthroline). C. Western blot analysis of 3XFLAG-Pch2 in cells (wild type or 3XFLAG-Pch2-E399Q) treated with mock or 1,10- Phenanthroline. D. ChIP-qPCR analysis of 3XFLAG-Pch2-E399Q at *PPR1* (primer pair: GV2390/yGV2391), *GAP1* (primer pair: GV2597/GV2598) in cells (wild type or 3XFLAG-Pch2-E399Q) treated with mock or 1,10- Phenanthroline. Error bars represent standard error of the mean of at least three biologically independent experiments performed in triplicate. E. Western blot analysis of Rpo21 (α-FRB or α-Rpo21) in wild type and *rpo21-FRB* anchor away strains, upon treatment with DMSO or rapamycin, treated as described in Fig 3C. F. Representative image of immunofluorescence of Hop1 and Rpo21 on meiotic chromosome spreads. G. Quantification of non-nucleolar Pch2 intensity per spread nucleus in cells

treated with DMSO or rapamycin for 90 minutes, according to the scheme indicated in Fig 3C. H. Representative image of immunofluorescence of Pch2 on meiotic chromosome spreads in *ndt80Δ* and *ndt80Δzip1Δ* cells, as quantified in 3H. I. Quantification of nucleolar Pch2 intensity per spread nucleus in cells treated with DMSO or rapamycin. n.s. (non-significant) indicates p>0.05, Mann-Whitney U test.
(TIFF)

**S6 Fig.** A. ChIP-qPCR analysis of ORC (α-ORC) along the *GAP1* locus during meiotic G2/ prophase (4 hours). Primers pair 1: GV2595/GV2596, 2: GV2597/GV2598, 3: GV2599/ GV2600. Error bars represent standard error of the mean of at least three biologically independent experiments performed in triplicate. B. Flow cytometric analysis of *ORC1* and *orc1-161* cells (wild type, or expressing or 3XHA-Pch2-E399Q). Time (hours) after induction into the meiotic program is indicated. Experiment was performed at 30˚C. C. ChIP-qPCR analysis of Orc2-TAP at *PPR1* (primer pair: GV2390/GV2391), *GAP1* (primer pair: GV2597/GV2598) and *ARS1116* (primer pair: GV2577/GV2578) in *ORC1* and *orc1-161* cells. Experiment was performed at 30˚C. Error bars represent standard error of the mean of at least three biologically independent experiments performed in triplicate. D. Western blot analysis of Orc1 and Orc2 (α-ORC) in *ORC1* and *orc1-161* cells. Experiment was performed at 30˚C. E. Western blot analysis of 3XHA-Pch2 (α-HA) in *ORC1* and *orc1-161* cells. Hours after induction into the meiotic program indicated. Experiment was performed at 30˚C. F. Gene expression of ORFs *PPR1* (primer pair GV2390/GV2391), *GAP1* (GV2597/GV2598), and *HOP1* (primer pair GV2605/GV2606) relative to β-Actin (*ACT1*; primer pair GV2717/GV2718) in *orc1-161* and *ORC1* cells grown at 30˚C. Relative gene expression of the reference strain (*ORC1*) was set to 1. G. Western blot analysis of Orc1 and Orc2 (α-ORC) in *ORC2* and *orc2-1* cells. Experiment was performed at 30˚C. H. Flow cytometric analysis of *ORC2* and *orc2-1* cells (wild type, or expressing or 3XHA-Pch2-E399Q). Hours after induction into the meiotic program indicated. Experiment is performed at 30˚C. I. ChIP-qPCR analysis of 3XHA-Pch2-E399Q at *PPR1* (primer pair: GV2390/GV2391) and *GAP1* (primer pair: GV2597/GV2598) in *ORC2* and *orc2-1* cells. Experiment was performed at 30˚C. Error bars represent standard error of the mean of at least three biologically independent experiments performed in triplicate.
(TIFF)

**S7 Fig.** A and B. Representative images of immunofluorescence of meiotic chromosome spreads in 3XHA-Pch2 expressing *ORC1* or *orc1-161* cells collect at 4 hours (A) or 5 hours (B) after induction into the meiotic program. Experiments were performed at 30˚C. For quantification see Fig 4G. C. Flow cytometric analysis of *ORC1* and *orc1-161* cells of cells that were analyzed in Fig 3F and 3G, and S7A and S7B Fig. Time (hours) after induction into the meiotic program is indicated. Experiment was performed at 30˚C. D. Western blot analysis of 3XHA-Pch2 and Orc1 (α-HA and α-ORC) in *ORC1* and *orc1Δbah* cells. Hours after induction into the meiotic program indicated. E. Flow cytometric analysis of *ORC1* and *orc1Δbah* cells (wild type, or expressing 3XHA-Pch2-E399Q). Hours after induction into the meiotic program indicated.
(TIFF)

**S8 Fig.** Hierarchically clustered heatmap based on correlation coefficients using from 3XFLAG-Pch2-wild type ChIP-seq datasets as inputs. Data for histone modifications are from [56]. Spearman correlation values are indicated.
(TIFF)

**S9 Fig.** A. Representative image of immunofluorescence of Hop1 and Gmc2 on meiotic chromosome spreads in *pch2Δndt80Δ* and *ndt80Δ* cells. B. Western blot analysis of Hop1 and Pch2

(α-HA) in *rpo21-FRB* anchor away *ndt80Δ* and *pch2Δndt80Δ* cells progressing synchronously through meiotic prophase, treated with DMSO or rapamycin, as indicated above the western blot image. Arrow indicates phosphorylated Hop1, * indicates non-phosphorylated Hop1. C. Quantification of total Hop1 intensity per spread nucleus after treatment as indicated. n.s. (non-significant) indicates p>0.05, Mann-Whitney U test. D. Schematic of treatment regimen used for anchor away experiment for which quantification is shown in Fig 5B. Representative immunofluorescence of meiotic chromosome spreads in the 3XHA-Pch2 expressing *rpo21-FRB* anchor away cells, treated with DMSO or rapamycin as indicated. Chromosome synapsis was assessed by α-Gmc2 staining. E. Representative images of Hop1 ChIP-seq binding patterns in wild type and pch2Δ from [44]. Chromosome coordinates and primer pairs are indicated. F. ChIP-qPCR analysis of Hop1 in *rpo21-FRB* anchor away strains, upon treatment with DMSO or rapamycin, treated as described in Fig 2C. Primers are described in [44]. Error bars represent standard error of the mean of at least three biologically independent experiments performed in triplicate. G. Representative images of immunofluorescence of Hop1 on meiotic chromosome spreads in *ORC1* or *orc1-161* cells collect at 4 hours after induction into the meiotic program. Experiments were performed at 30˚C. Chromosome synapsis was assessed by α-Gmc2 staining. *Indicates nucleolar region. H. Quantification of total Hop1 intensity per spread nucleus (as shown in S4G Fig) after treatment as indicated. n.s. (non-significant) indicates p>0.05, Mann-Whitney U test.
(TIFF)

**S10 Fig.** A. ChIP-qPCR analysis of 3XHA-Pch2-E399Q at *PPR1* (primer pair: GV2390/GV2391), *GAP1* (primer pair: GV2597/GV2598) during meiosis, and of *PPR1* (primer pair: GV2390/GV2391), *GAP1* (primer pair: GV2597/GV2598) and *TDH3* (primer pair: GV2591/GV2592) and *DAL4* primer pair: GV2601/GV2602) during mitosis. Time (hours) is indicated. Error bars represent standard error of the mean of at least three biologically independent experiments performed in triplicate. B. Western blot analysis of 3XFLAG-Pch2 and Zip1 (α-FLAG and α-Zip1) in wild type and *zip1Δ* cells. Time (hours) after induction into the meiotic program is indicated. C. Flow cytometric analysis of wild type and *zip1Δ* cells (wild type, or expressing 3XFLAG-Pch2-E399Q). Time (hours) after induction into the meiotic program is indicated. D. mRNA quantification of *GAP1*(primer pair: GV2597/GV2598) *(primer pair: GV2717/GV2718) or *PPR1* (primer pair: GV2390/GV2391)/β-Actin (*ACT1;* primer pair: GV2747/GV2748) in wild type or *zip1Δ* cells during meiotic G2/prophase (4 hours).
(TIFF)

**S11 Fig.** A. Pearson's correlation analysis between expression levels (log$_2$ TPM) of *ndt80Δ* and *pch2Δndt80Δ*. B. RNA seq-differential expression values from *pch2Δndt80Δ* relative to *ndt80Δ*, for all genes, wild type Pch2-binding and KEGG pathway (meiosis) genes. C. Histogram depicting the Log$_2$ normal distribution of averaged TPM values from ndt80 strains RNA-seq. The defined expression cut-off (red line, see methods) and the range of TPM of 3X-FLAG-Pch2-wild type binding genes (green line) are indicated.
(TIFF)

**S1 Table. List of Pch2 binding CDS sites.**
(XLSX)

**S2 Table. Gene Ontology (GO) terms of Pch2 binding ORFs.**
(XLSX)

**S3 Table. Underlying numerical data of main results.**
(XLSX)

**S4 Table. Western blot quantification of selected blots presented in the manuscript.**
(XLSX)

**S5 Table. Differential expression analysis (log2 fold) of *ndt80Δpch2Δ* relative to *ndt80Δ* cells.**
(XLSX)

## Acknowledgments

We thank Arnaud Rondelet, Vivek B. Raina (both from Max Planck Institute of Molecular Physiology, Dortmund, Germany) and John Weir (Friedrich Miescher Laboratory, Tübingen, Germany) for comments on the manuscript and members of the Vader and Bird laboratories for ideas and helpful discussions. We thank Andrea Musacchio (Max Planck Institute of Molecular Physiology, Dortmund, Germany) for ongoing support. We acknowledge Stephen Bell (MIT, Cambridge, USA), Amy MacQueen (Wesleyan University, Middletown, USA) and Nancy Hollingsworth (Stony Brook University, Stony Brook, USA) for sharing reagents.

## Author Contributions

**Conceptualization:** Richard Cardoso da Silva, Gerben Vader.

**Data curation:** Richard Cardoso da Silva.

**Formal analysis:** Richard Cardoso da Silva.

**Funding acquisition:** Richard Cardoso da Silva, Gerben Vader.

**Investigation:** Richard Cardoso da Silva, María Ascensión Villar-Fernández.

**Methodology:** Richard Cardoso da Silva.

**Supervision:** Gerben Vader.

**Visualization:** Richard Cardoso da Silva, Gerben Vader.

**Writing – original draft:** Richard Cardoso da Silva, Gerben Vader.

**Writing – review & editing:** Richard Cardoso da Silva, María Ascensión Villar-Fernández, Gerben Vader.

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
