## [Decision Letter · Decision Letter 0]

13 Nov 2019

Dear Gerben,

Thank you very much for submitting your Research Article entitled 'Active transcription and Orc1 drive chromatin association of the AAA+ ATPase Pch2 during meiotic G2/prophase' to PLOS Genetics. Your manuscript was fully evaluated at the editorial level and by independent peer reviewers. The reviewers appreciated the attention to an important problem, but raised substantial concerns about the current manuscript. Based on the reviews, we will not be able to accept this version of the manuscript, but we would be willing to review a much-revised version. We cannot, of course, promise publication at that time.

Reviews are given at the bottom of this letter; however, I (ML) would like to emphasize two points in particular:

1. A major concern that I (ML) had, and that reviewers expressed, involves the use of "peak shape score" as a proxy for occupancy in the analysis of ChIP-seq data. As far as I can tell (and the reference describing this metric is pretty opaque on the issue), this is a metric that can be independent of actual occupancy levels--for example, a sharp peak in a region of low background occupancy could have a higher score than a broad peak of higher occupancy levels in a region of high background occupancy. My suspicion is that differences in actual occupancy profiles in the ChIP-seq experiments (i.e. ChIP/input) are much smaller, and it is of course this value, not one reflecting peakiness, that is of actual biological relevance. Given this concern, I think that it is important to provide corresponding ChIP-seq profiles that report ChIP/input ratios for the regions examined (e.g. Figure 1C, D, Fig 4A, Fig S5, Fig S6F) and to provide ChIP-qPCR values for primer pairs at peak loci (i.e. GAP1, HOP1, etc) not only within the "peak" area but also outside them. I think that it is likely that the reduction in Pch2 occupancy seen under conditions of Pol2 inhibition/depletion and of ORC mutants are going to turn out to be regional rather than gene body-specific, but of course that remains to be seen. I believe that this concern is also what underlies reviewers' requests for additional ChIP-seq data. It must be emphasized that this is a critical concern; until true occupancy data are in hand, it will not be possible to evaluate the biological significance of the experiments. 

2. A second concern involves the experiments with *orc* mutants. In general, the defect in Pch2 occupancy seems to parallel defects in replication; *orc1-161* and *orc1-∆BAH* show delayed replication, while *orc2-1* does not. It seems quite possible that, at the 4h time point when ChIP is done, some aspect of meiotic chromosome structure is incompletely established in mutants where replication is compromised, and this, rather than a direct Orc1 interaction, is responsible for the observed effects. Unless the Pch2-Orc1 interaction can be disrupted without affecting replication dynamics, this will remain an open question, and the conclusions regarding the contribution of Orc to Pch2 recruitment will need to be substantially revised.

Further editorial comments are given below immediately before the reviewers' comments. 

Should you decide to revise the manuscript for further consideration here, your revisions should address the specific points made by each reviewer and the editor. We will also require a detailed list of your responses to the review comments and a description of the changes you have made in the manuscript. I apologize for the absence of numbers in comments from reviewers 1 and 3--please fell free to insert numbers as you see fit if it will help in crafting your response. 

If you decide to revise the manuscript for further consideration at PLOS Genetics, please aim to resubmit within the next 60 days, unless it will take extra time to address the concerns of the reviewers, in which case we would appreciate an expected resubmission date by email to plosgenetics@plos.org. 

[LINK]

We are sorry that we cannot be more positive about your manuscript at this stage. Please do not hesitate to contact us if you have any concerns or questions.

Yours sincerely,

Michael Lichten, Ph.D.

Associate Editor

PLOS Genetics

Gregory P. Copenhaver

Editor-in-Chief

PLOS Genetics

AE's additional comments:

3. (comments 1 and 2 are in letter body, above). PLOS has a very strict data availability policy (please see letter body, above and journals.plos.org/plosgenetics/s/data-availability). In particular, the term "data not shown" is verboten; either show data or revise the text so that the statement is not necessary.

a. Relevant lines in the manuscript are 157, 160-4, 252, 254, 329 and 392; for line 153, I suggest that you simply say that the current paper describes Pch2 occupancy patterns in non-rDNA sequence, and that binding patterns in rDNA will be described elsewhere.

b. The ChIP-seq data need to be made available to reviewers, so please give the password in manuscript on line 521--reviewers do not have access to the cover letter. 

c. For all figure panels with graphs, please provide the data underlying each graph. This can most easily be by including an Excel workbook with a separate worksheet for each panel. For example, for Figure 1F, please include all values for each primer pair in both tagged and untagged strains. (This need not be done with very data-dense panels such as 1C, D or E).

4. Figure legends--I assume that by “standard error” you mean standard error of the mean; please clarify, and also please clarify what “at least three independent experiments performed in triplicate” means—and how SEM was calculated. Were there three sets of three values, or just three values?

5. Fig S1C—correlation seems to be driven entirely by genes with no expression. Once these are eliminated, the is probably not correlation at all.

6. line 67: Red1 shares no homology to SYCP2/3; better to make clear that the proteins are functional analogs but do not share significant homology.

7. line 137: Capitalize Materials and Methods; please use 10(superscript)-15 rather than 1e-15

8. Figure 1F—given the apparent absence of Pch2 from sequences up- and down-stream of GAP1, it would seem a good idea to include qPCR primers for these regions, as well; this would be a good control for the Pol2 ChIP as well (see AE comment 2, above).

9. Figure 1G—please quantify FLAG/Pgk1 ratios; it appears that Pch2 243-564 is present at substantially lower levels. Is Pch2 243-564 localized to the nucleus?

10. Other western blots—please quantify and present ratios for the following; Figure 1B, 2G, 4F (including upshifted Hop1), 5B, S4B, S4E, S4F, S4I. For Figure S7C, it will be important to have the Zip1 and zip1∆ samples on the same blot, to determine whether or not Pch2-E399Q levels are similar in both strains. Ratios can be reported under each lane, or, if it makes figures too crowded, in a supplementary table.

11. Figure S1B—why is RNA-seq under High, Medium and Low, while mRNA levels is under all 4?

12. Figure S3A—was the phenanthroline treatment here as shown in panel B?

13. Figure S4F—at what time in meiosis were these protein samples taken?

14. Figure S6F—please provide Y axis labels and values. I could find no such images in Subramanian et al, and am assuming that you re-analyzed their data. Please provide full information for this analysis, including the datasets analyzed, method used, etc. 

15. Please proofread references and correct title capitalization, italicization of genes and species, journal abbreviations, etc. 

16. Please carefully read the instructions to authors (https://journals.plos.org/plosgenetics/s/submission-guidelines) and follow formatting instructions (helpful examples can be downloaded) in crafting the revision. 

Reviewer's Responses to Questions

**Comments to the Authors:**

Reviewer #1: Notes on Pch2 PLoS Genetics Manuscript

In this manuscript, Cardoso da Silva et al. examine the meiotic chromosome localization of Pch2, a highly conserved and important regulator of meiotic recombination. The key experiment here - Pch2 ChIP-Seq - is of high interest to the field and one that is potentially very informative about Pch2's regulation and functions. It has the potential to help us understand Pch2's seemingly diverse roles: in controlling the localization of the recombination regulator Hop1 on chromosomes, in suppressing recombination in the repetitive rDNA, and in its mysterious checkpoint-like function in mutants with compromised inter-homolog synapsis.

While the topic is of high relevance and the data interesting, I can't recommend publication without significant revisions to address a number of major and minor questions/concerns, which are listed in detail below. In summary, while the authors have identified a set of Pch2-localizing sites on chromosomes in meiotic prophase, they do not adequately explain how Pch2 gets to these sites (yes, it's transcription and Orc1-dependent, but why?), nor do they reveal any biological role for these localization sites. That is, the sites identified do not seem relevant to Pch2's most well-recognized function, in Hop1 remodeling/removal; this may be a consequence of the depletions etc. being done too late in meiotic prophase to see a difference. The authors also inexplicably fail to present some seemingly very relevant data and analyses, noting them as "unpublished observations": this includes data on Pch2's localization to rDNA, which would arguably be one of the most interesting features of their ChIP-Seq dataset, and an analysis of Pch2 ChIP-Seq compared to various histone modifications. I am also somewhat concerned about the peak-calling algorithm, and features of some figures that suggest that the observed Pch2 localization is very weak.

Finally, I hate to suggest excessive extra work in a review, preferring instead to review the material as presented. Nonetheless I can't help but note that the manuscript would be much stronger if ChIP-Seq had been performed instead of ChIP-qPCR for many comparisons. This would allow genome-wide correlations of, e.g., Pch2 and Orc1 localization in meiotic prophase, Pch2 localization in mitotic cells, etc. etc. On a similar note, because meiotic prophase stage is so very important in understanding Pch2 function, a time-course for Pch2 localization (at selected sites by ChIP-qPCR or better, by ChIP-Seq) would be extremely informative.

Major comments/concerns:

Page 6 - The authors do most of their assays four hours after induction of meiosis. What stage of meiotic prophase are these cells in? I can imagine that catching Pch2 in its (presumably) most physiologically-relevant state - during synapsis, when it is localizing to chromosomes and remodeling/removing Hop1 from the axis - might be difficult to accomplish. If the cells are pre-zygotene or already in pachytene by this analysis, what is the relevance of the localization the authors are mapping?

Page 6 - Please describe "peak shape score" in more detail. From my quick literature search, this seems to be a score generated by a robust "learning" peak-calling method, but it bears some explanation in the text. For example, what is a "good" peak shape score?

Page 6 - The authors note (several times) the fact that they observed Pch2 localization to the rDNA by ChIP, yet choose not to describe it in this manuscript. I can't see why it's excluded; it seems very relevant to our understanding of how Pch2 works.

Figure 1E does not make sense to me. If the authors are averaging the ChIP-Seq signal over ~500 peaks, why is the shape of this graph so jagged, instead of smooth? Also, why is the signal seemingly so low? It seems from the graph that the baseline level is 0.8 ChIP-Seq/Input (log2 ratio), while the peaks are only 0.92? Can this be right? The peaks are only 1.9-fold enriched over background binding, and the background is somehow 1.75-fold enriched over background? What's going on here? If these are the real ratios, is the peak-calling algorithm just enhancing noise? Figure 3B is also much more jagged than expected from an average of many peaks.

Figure 2I, 3G, and (maybe) 4D) - It seems that counting the number of foci of Pch2 (Hop1 in Figure 4D) is not really the right way to quantify the data. There is obviously a dramatic reduction in overall intensity of chromosome-associated Pch2 in Rapamycin treatment and orc1-161 cells (assuming the panels are intensity-matched), but the number of foci changes much less than the overall intensity. Is there are way to quantify overall Pch2 intensity per nucleus, rather than number of foci?

Is ORC localized to origins of replication in meiotic prophase? If not, Pch2 localization there would not be expected.

The authors cite a work by Shor et al. (ref. #45) that identified a large number of ORC-binding sites in highly-transcribed genes close to replication origins. Have the authors compared this set of genes to their own identified sites? TDH3, UTH1, and SSA1 are on the list of previously-identified "ORF-ORC sites" (not GAP1, however). The meiotic genes would not have been identified in that prior analysis, but it still may be a worthwhile comparison.

On the subject of the above comment, it seems that Shor et al. showed that the "ORF-ORC" sites are all pretty close to origins. Have the authors looked at the proximity of their identified Pch2 peaks to origins, comparing that proximity to that of the typical gene to an origin?

One more comment on the above, given the authors' experiments with Pch2 expression in mitosis seems to have shown no localization with GAP1, what about analyzing other sites? It seems that the authors should at minimum examine several other loci, or ideally perform ChIP-Seq again on these cells to compare all sites. It could be that GAP1 is atypical of sites (and HOP1 is not transcribed in mitosis - this is a fine negative control, but doesn't invalidate this point).

Regarding the authors' result that deletion of the Orc1 BAH domain compromises Pch2 localization to gene bodies: Orc1's BAH domain is known to localize to H3K20 methyl marks, at least in other species. Can the authors comment on the possibility that the peaks they identify might be enriched in this mark, perhaps specifically in meiosis, for some reason?

Figure 4 - The experiment measuring Hop1 localization to chromosomes when disrupting Pch2 localization to gene bodies needs to be explained better. The interpretation of this data depends very strongly on what stage the cells are in during these measurements: If they are in leptotene/zygoetene, the authors can reasonably argue that the Hop1 they are measuring is the population of Hop1 that is promoting DSBs and COs prior to synapsis. If the cells are in pachytene during this analysis, then the authors are only looking at the "residual" Hop1 that remains on synapsed chromosomes, and is presumably not (or only weakly) promoting DSBs/COs. In the same vein, also matters a lot how long Pch2 localization has been compromised (i.e. if the Rapamycin was added after synapsis and the completion of Pch2's Hop1-removal task, then its depletion from chromosomes would not be expected to affect Hop1 levels.

Related to the above point: Lines 333 and 334, the authors state that PNAPII depletion did not cause differences in the amount of phosphorlyated Hop1 (Figure 4E-F). But again this experiment was done entirely in pachytene, after the bulk of phosphorylated Hop1 had been removed/degraded/dephosphorylated. Depleting RNAPII during the meiotic prophase stage when Pch2 activity is most relevant for Hop1 (zygotene) would seem to be more informative.

Discussion - the authors note that the Orc1 BAH domain may in fact associate with Dot1-mediated H3K79 methylated nucleosomes, and state that they detected a correlation between Pch2 ChIP-Seq and H3K79 methylation; but do not show this data. Why not? I think an important supplemental figure would be to show a correlation analysis of Pch2 ChIP-Seq peaks with every histone modification for which a dataset is available. Are any such datasets available from meiotic cells?

When discussing the loop-axis structure of meiotic chromosomes, the authors should cite two recent Hi-C papers by Muller...Koszul (Mol Syst Biol 14(7):e8293) and Schalbetter...Neale (Nature Communications 10(1):4795, October 2019) describing the yeast loop-axis structure in detail, and perhaps also the mammalian-cell Hi-C papers by Alavattam et al (NSMB 2019), Wang et al. (Mol Cell 2019), Patel et al. (NSMB 2019), and Covadonga Vara et al. (Cell Reports 2019)

Minor points:

Figure 1F - this would be much more useful if the authors added one or a couple probes outside the ChIP-Seq peak for GAP1.

Line 148: Reference Supplementary Fig. 1A.

Line 185-188: Please double check with the figure reference.

Line 234 - "PolI" should be "PolII"

Reviewer #2: This work by Cardoso da Silva et al. reports the association of Pch2 with a subset of chromosomal regions containing genes that are actively transcribed by RNA polymerase II during meiosis. In fact, active transcription is required for Pch2 recruitment to those sites. Paralleling the previously known role of Orc1 (a subunit of ORC) in Pch2 localization to the rDNA, this study shows that Orc1, but not Orc2, is also required for Pch2 binding to selected gene bodies in euchromatin. However, unlike ORC, Pch2 is not detected at replication origins. In addition, Zip1 is also needed for Pch2 association to transcribed regions. However, impairment of Pch2 chromatin recruitment by depletion of the Rpo21 component of RNAPII does not result in any effect in Hop1 chromosomal distribution and abundance, suggesting that the chromosomal population of Pch2 described in this work is not involved in regulating Hop1 function.

Given the importance of Pch2 in the control of various aspects of meiotic chromosome metabolism, the findings reported in this manuscript are interesting in the meiosis field. The experiments performed in this work are very well executed, the results are clear and, for the most part, support the conclusions presented. The main weakness of this paper is that it does not provide any insight into the possible biological relevance for meiosis of the transcription-associated population of Pch2 reported here.

Major comments/concerns

1) Immunolocalization of Rpo21-FRB on chromosome spreads in the absence of rapamycin (+DMSO) shows a rather exclusive pattern with that of Zip1 in the single nucleus presented in Figure 2D. Curiously, this pattern is reminiscent of Hop1 localization, also displaying an alternative distribution with Zip1 on wild-type prophase I chromosomes (San-Segundo and Roeder, Cell 1999; Joshi et al., PLoS Genetics 2009). The authors could test whether Rpo21 foci, representing transcription sites, and Hop1 colocalize.

2) Figures 2H, 2I (as well as Figures S6B, S6C) conclude that depletion of Rpo21 leads to fewer Pch2 chromosomal foci without affecting nucleolar Pch2. From the representative nuclei shown, it appears that, upon Rpo21 depletion, Pch2 foci are not only fewer, but also weaker. This raises the question of whether the remaining foci represent the same population of Pch2 that depends on transcription with lower amount of Pch2 loaded or, alternatively, that those Pch2 remaining foci correspond to an Rpo21-independent population and, therefore, may be involved in Hop1 chromosome exclusion. In other words, since by ChIP-seq only one chromosomal fraction of Pch2 is detected, which is transcription-dependent and not involved in Hop1 distribution, do the cytological assays detect only this same population? First, quantification of the Pch2 nucleolar signal relative to chromosomal Pch2 must be performed to definitely state that the nucleolar pool is unaffected. Second, colocalization between Hop1 and Pch2 before and after Rpo21 depletion must be assessed and quantified to determine if the population of Pch2 that remains after transcription inhibition shows a different chromosomal distribution relative to Hop1 or there is some degree of overlap. Also colocalization between Gmc2 and Pch2 (-/+ Rapamycin) should be quantified.

3) This same issue applies to Figures 3F, 3G, where Pch2 localization is analyzed in the orc1-161 mutant. How is Hop1 distribution/abundance in the orc1-161 mutant? Is Orc1 only required for recruitment of the transcription-associated chromosomal Pch2 pool not involved in Hop1 removal from the SC axes, or it also affects the other proposed population of Pch2 presumably acting on Hop1?

4) The authors view as “striking” the observation that Pch2 ChIP-seq peaks do not coincide with Hop1-enriched (axis) sites. Given the assumption that Pch2 removes Hop1 from chromosomes, I would consider that result as “expected”. Another question is the finding that the ATP-hydrolysis defective Pch2-E399Q version neither shows overlap with Hop1 regions; this is more surprising. As the authors explain in the manuscript, Pch2-E399Q is expected to remain locked to its client(s)/substrate(s)/adaptor(s). Indeed, according to this notion, cytological assays have shown that, in contrast to wild-type Pch2, Pch2-E399Q extensively colocalizes with Hop1 both at the rDNA and chromosomes (Herruzo et al., NAR 2016). Consistent with the prediction for a more stable interaction with the substrates, Pch2-E399Q displays stronger chromatin binding compared to Pch2 as shown in Figure S1A. The authors speculate that there are two chromosomal populations of Pch2, one associated to certain transcribed regions and another one, undetectable by ChIP, responsible for Hop1 removal. Why this second pool cannot be detected, at least in the case of Pch2-E399Q, if it remains locked to Hop1, and Hop1 is detectable? What is the direct evidence that this second population of chromosomal Pch2 does exist? Do the cytologically detectable Pch2 foci represent both populations? (Please, see points 2 and 3).

5) The authors speculate that chromatin modifications may influence Pch2 binding to certain transcribed genes. Since Sir2 and Dot1 are the usual suspects in this context, it would be interesting to determine whether these chromatin modifiers are required for Pch2 recruitment to gene bodies.

6) The finding that Pch2 binds to a subset of transcribed genes during meiosis, raises the possibility of a possible role for Pch2 in regulating transcription that is not addressed in the manuscript. It is curious that one of the genes bound by Pch2 is HOP1. It has been shown that global levels of the Hop1 protein are increased in the pch2 mutant (Ho and Burgess, PLoS Genetics 2011), although the underlying molecular mechanism is currently unknown. Figure 4F shows that Hop1 protein levels and Hop1 phosphorylation do not appear to be significantly affected upon Rpo21-FRB depletion, a condition that prevents Pch2 binding to transcribed genes. However, quantification and repeats of this WB analysis, including pch2 deletion as control for comparison, should be provided. Also, since some low amount of Pch2 bound to genes may remain in the Rpo1-FRB depletion assay, to definitely rule out a role for Pch2 in HOP1 transcription, HOP1 mRNA levels in the pch2 deletion mutant should be analyzed.

Minor points:

1) Figures 1G and 1H show that elimination of the NTD of Pch2 impairs Pch2 binding to selected target genes. Thus, the authors conclude that the NTD is required for Pch2 recruitment to transcribed gene bodies. Although, the conclusion is formally correct, it is possible that the NTD is not directly involved in recruitment, but in the formation or stability of the Pch2 hexameric complex.

2) What do the asterisks indicate in Figures 2H and S6B? The rDNA? Please, explain in the legends

3) Evidence for functionality of the 3xFLAG tagged version of Pch2 must be provided

4) In the WB of Figure 1B the Pch2 protein is detected at t=0 using anti-FLAG antibodies, but not in Sup Fig S4E using anti-HA antibodies? Is that an antibody issue?

5) Line 148: Supplementary Figure 1B should be 1A

6) Line 168: Supplementary Figure 1C should be 1B

7) Line 171: Supplementary Figure 1D should be 1C

8) Line 185: Supplementary Figure 1E should be 1D

9) Line 188: Supplementary Figure 1F should be 1E

10) In page 1 of Supplementary data, Supplementary Figure 1D should be 1C

11) In page 2 of Supplementary data, Supplementary Figure 1B should be 1A

Reviewer #3: Overview.

Here the authors have used whole-genome analysis and genetics to investigate the role of the the AAA+ ATPase Pch2 during meiosis in S. cerevisiae, a widely employed model system for studying the mechanisms of recombination and chromosome segregation. The authors identify regions of enriched binding along chromosomes and draw connections to the locations of genes, which is a very different pattern of enrichment to that observed for axis-associated proteins like Rec8 and Hop1. Connections with active transcription are not convincing, and it is not clear that the point the authors are trying to make is well supported. Initial data use whole-genome maps, but most of the data is targeted Q-PCR at a small number of loci. Given the differences across loci that the NGS ChIP-seq data reveals, it is not clear how representative the Q-PCR loci are. ChIP-seq in other mutants (rather than just Q-PCR) would certainly strengthen the validity of the conclusions. From a technical standpoint, I am not convinced by the use of "Peak shape score" to plot (and analyse?) raw ChIP-seq data. Is this a quantitative measure of enrichment? Or just a measure of how "good" a peak there is in the data? I offer below a number of comments that I think will help to strengthen and improve the clarity of the presented research.

General comment.

The text would benefit greatly from inserting more paragraph breaks and, ideally, section headings so that the logical flow from one idea/figure to the next is clear to the reader. The current text is presented, largely, as one continuous body of text that is hard to traverse.

Specific comments.

Line 131-132. Where are the individual datasets presented? Where is the demonstration that they are highly correlated? I suggest the authors include a correlation between each of the binned datasets. Plotting, as overlays, the individual datasets along part of a chromoosme may also help them make this point.

Line 135. The rationale for the manner in which data are processed and peaks are called is not well described nor justified. This leads to a number of questions that require clarification:

What does "Peak shape score", the Y-axis of plots in Fig 1C, 1D (and elsewhere) mean?

Is it actually a quantitative measure of binding, or just a measure of how well (confident?) the algorithm identified a peak?

What if most of the signal is dispersed, and the real signal doesn't inherently have peaks?

Do peaks represent the majority, or the minority, of the signal reported?

the use of a very stringent threshold for peak-calling raises alarm bells to this reviewer...how biased is the subset that are being analysed to only those regions with a particular "shape" and or "strength"?

What do the data look like if plotted as a more traditional signal/input?

Line 138. The fact that the peaks arise in only a subset of RNAP PolII genes is hardly surprising, when the thresholding used means that there are only a (relative) handful of peaks being analysed...is this a fair description of the data?

Line 140...The statement that 3% of peaks arose elsewhere is somewhat misleading. The question in this reviewer's mind is: How much of the enriched signal was present within the called peaks?

Fig 1C. Plotting the locations of ORFs would be helpful to see a wider trend of correspondence.

Fig 1D. Equally, plotting a wider area than displayed in Fig 1D would aid the comparison. More importantly form this plot it is not clear what the message is. Are all these genes enriched ? (They seem to be.) Clarifying the description of these data would greatly help the reader.

Line 143 and Fig 1E. The analysis of correspondence with gene structure (and absenece in promoters) would be strengthened by a figure similar to 1E, but where the analysis/pileup point is the gene promoter (and/or terminator). The two plots together would help to demonstrate the enrichment within ORFS and (stated, but not demonstrated) exclusion from promoters.

Labelling for Sup Fig 1 seems to have got mixed up. I think all the panels are one letter out.

Figure referencing used below is based on the submitted figure labelling, not the text.

Fig S1A. How many peaks were present in both samples? A correlation of the shared set of peaks would be informative to determine if, on average, the same peaks are stronger in the mutant (or variable?), or whether the global increase reported is because they represent different peaks detected in the mutant.

Line 163. Where is the data shown to support this statement that all genes occupied by Pch2 are transcribed during meiosis? What (arbitrary) cut-off was used to determine which genes are meiotically transcribed? Is it to do with induction? Or absolute level? These data and conclusions are not currently convincing.

Indeed, Fig S1B, C seems to directly indicate that Pch2-bound genes span the entire spectrum of gene expression levels.

Moreover, Line 169, requires revision: Fig 1C is about as close to no correlation (r^2=0.0026) as one can get... Why, then does the text state that a weak correlation was detected? The authors may wish to compute the correlation they (may) observe between the same number of random values for comparison...

Separately, is it known whether "Peak shape score" is a quantitative measure? Perhaps there is a correlation between fold enrichment and expression, but not one between "Peak shape score" and expression, for example.

Line 173-179. The experiments performed by the authors to try to rule out ChIP artefacts are good, but certainly require a more thorough summary in the main text. The authors point out where artefacts may arise, so quite simply they then need to succinctly state (in the main text) how they are able to rule them out systematically. This will strengthen, not weaken, the main text.

Fig S3A. Minor. Is 20% EtOH, the final concentration? What concentration is the drug?

Fig S3B. Minor. I suggest placing this schematic before Fig S3A.

Line 231. It would strengthen the conclusion if the published zip1D experiments (reported to show complete loss of Pch2) were repeated alongside the new experiments reported here for RNA Pol inhibition or depletion. i.e. Is it possible that this is a lab/reagent/analysis difference rather than a real difference? If this cannot be done, a qualification can be included to this effect.

Line 232. It is perplexing why the authors hypothesise the effect may indicate incomplete inhibition when the data presented directly demonstrate this to be the case (Fig. 2E).

Line 250-252. Please clarify this point. In the previous sentence the fold enrichment of Pch2 is assayed around 798 annotated ARSs (nothing to do with Pch2 peaks). Why then would adjusting the p-value threshold for Pch2 peaks make any difference? Are the authors themselves clear about what analyses they have performed?

Lines 303-310 and Fig 4A. Without further description and validation, I remain sceptical about plotting data Y-axis as "Peak shape score". Is this a known quantitative value that measure enrichment of these factors? Can these data be plotted as more simple fold enrichment instead? Also, an X-Y plot of binned Pch2 enriched signal vs the other factors would help to understand if this is a global quantitative positive, negative, or wholly absent relationship.

Line 328. Under ndt80D conditions, isn't most Hop1 already lost on chromosomes? Is there dynamic range to observe an effect?

Would performing anchor-away Pch2 be helpful here (positive control) perhaps?

Lines 329-330. This part of the sentence is confusing. Perhaps clarify that loci used for analysis of Hop1 were identified based on prior datasets.

Lines 331-332. Please clarify what the hypothesis is here with respect to investigating Hop1 phosphorylation state.

**Have all data underlying the figures and results presented in the manuscript been provided?**

Reviewer #1: No: ChIP-Seq data should eventually be deposited somewhere

Reviewer #2: Yes

Reviewer #3: No: The Pch2 dataset is not currently available publicly: https://www.ncbi.nlm.nih.gov/geo/query/acc.cgi?acc=GSE138429

PLOS authors have the option to publish the peer review history of their article (what does this mean?). If published, this will include your full peer review and any attached files.

Reviewer #1: No

Reviewer #2: No

Reviewer #3: No

---

## [Decision Letter · Decision Letter 1]

12 Mar 2020

Dear Gerben,

Thank you very much for submitting your Research Article entitled 'Active transcription and Orc1 drive chromatin association of the AAA+ ATPase Pch2 during meiotic G2/prophase' to PLOS Genetics. Your manuscript was fully evaluated at the editorial level and by independent peer reviewers. The reviewers appreciated the attention to an important topic but identified some aspects of the manuscript that should be improved.

Below, some commentary on the reviewers’ comments:

Reviewer 1: these should be easily addressed

Reviewer 2: This reviewer had three major concerns.

Biological significance. I think that this is a concern that should be addressed in the discussion. For the difference between your conclusions and those in reference 39, I think that a bit more explicit discussion of what the differences in experimental approaches were, and why they might produce a difference in results, would be warranted. A quick read of that paper indicates that, as you say, there was very little non-nucleolar Pch2 detected (I think that this might be due to how images were thresholded, but I leave it to you to decided); also, ref 39 used a functional readout for Pch2 activity (checkpoint activation in zip1∆) that might actually be the difference between SK1 and BR, since BR zip1∆ mutants arrest and SK1 only delays. Again, I think that a more explicit discussion of this and other issues is warranted.With regards to the main concern, that of biological relevance, both your data and the data in reference 39 indicate that loss of Orc1 function does not have the same phenotype as a pch2 mutant, both with regards to Hop1 distributions and checkpoint function. The question remains then as to what the biological function is of Orc1-dependent Pch2 recruitment to transcriptionally active regions, and it would be useful to include at least some thoughts as to what function that recruitment might serve.  This concern involved the relationship between transcription and Orc1 and Pch2 recruitment. I agree with the reviewer that a trivial explanation would be that Orc1 is necessary for transcription of genes to which Pch2 is recruited. This should be relatively easy to address, either by looking at a couple of Pch2-marked genes, or by seeing if any data exists in the literature regarding the effect of orc1 mutants on gene expression. The question of whether or not transcription is required for Orc1 binding to these genes could, to my mind, be addressed by discussing the issue in an even-handed manner. The reviewer had concerns about the apparent contradiction between the correlation between Pch2 and H3K79me1, on one hand, and biological evidence that Dot1 limits Pch2 localization to euchromatin. Given the rather weak nature of the Pch2-H3K70me correlation, I think that it remains to be shown if this correlation reflects a direct or indirect effect, and I think that it would be wise to somewhat moderate the conclusions in this section. 

Reviewer 3: All the concerns of this reviewer need to be addressed, but it appears to me that most can readily be addressed by providing additional information in the text or in supplement without additional experimentation. However, I do wish to amplify the reviewer’s point regarding Supplementary Figure 1F, where it is abundantly clear that the regression line is being driven almost entirely by the points with expression values of ~0.8, and that removal of this subset would result in no correlation at all. In light of this, I would suggest that the conclusions be altered to say that transcriptional strength is not a factor that determines Pch2 binding.

Finally, this reviewer suggested (in their first point #1)(!) dividing the text further into paragraphs. I agree that this will increase readability and suggest the following lines as dividing points, to be divided after the full stop unless otherwise indicated. It would help if paragraphs were indented throughout the manuscript.

Line 134, at start

Line 154, at start

Line 170

Line 237

Line 257

Line 315

Also, please consider the following:

Line 331, suggest “we provide evidence below”

Line 395, suggest “also transcriptionally active”

Line 529-30, redundant with Lines 525-6

Throughout Materials and Methods, please provide antibody dilutions used.

References—improved, but journal abbreviations are still inconsistent.

Figures—for ChIP-seq graphs, please indicate bin sizes and provide units for “normalized reads” (i.e. such as reads per million per 10kb) in either the figure itself or the legend. 

We ask you to modify the manuscript according to the review recommendations before we can consider your manuscript for acceptance. Your revisions should address the specific points made by each reviewer.

[LINK]

Yours sincerely,

Michael Lichten, Ph.D.

Associate Editor

PLOS Genetics

Gregory P. Copenhaver

Editor-in-Chief

PLOS Genetics

Reviewer's Responses to Questions

**Comments to the Authors:**

Reviewer #1: The authors have done a good job addressing the most pressing and substantive questions and concerns brought up by all the reviewers. While it's still a mystery why Pch2 localizes to a subset of transcribed genes in meiosis, I am convinced that it does indeed do so. I just have a couple of minor points for the authors to address before acceptance:

Phenanthroline is mis-spelled twice, I believe.

Figure 2B - labels along the top are in the opposite order from what they should be.

Figure 6E - it’s a little confusing to have “zip1d” on a different line than “3XFLAG-Pch2-E399Q” - can the authors move it to the same line?

Reviewer #2: In this revised version of the manuscript by Cardoso de Silva et al. the authors have performed additional work and they have satisfactorily addressed and/or responded to most of the concerns raised to the previous version. However, there are still some issues that, in my opinion, remain to be solved.

1. Although the discussion regarding the possible existence of different populations of Pch2 with different functions (one presumably undetectable by ChIP-seq affecting Hop1 and another one detectable by ChIP-seq with no effect on Hop1, but both of them cytologically detectable) has been softened and additional possibilities are now mentioned, my main concern with this article still is the lack of some biological relevance for the transcription-dependent binding of Pch2 to chromosomes.

2. The authors now succinctly present some ChIP-seq data on the binding of Pch2 to the rDNA region describing that it binds to the RNAPI-transcribed 25S region. Since Orc1 is also required for Pch2 nucleolar targeting there could be also a connection between Orc1 and transcription by RNAPI in the rDNA. I agree that the detailed characterization of the Pch2 rDNA targeting is beyond the scope of this paper, but the possible connection between Orc1 and transcription on the “euchromatin” could be further explored. This paper describes that Pch2 recruitment to chromosomes requires Orc1 and requires RNAPII transcription, but the dependence/relationship between Orc1 and transcription is not analyzed. In other words, is Orc1 required for transcription at the ORFs bound by Pch2? Is transcription required for Orc1 binding to those ORFs? I don’t really like bringing up new issues during revision, but the results presented in the revised version lead to these questions that should be addressed or, at least, further discussed.

3. The authors now present a correlation analysis between genomic distribution of Pch2 and a panel of histone modifications obtained from mitotic cells. They comment on a certain degree of correlation between Pch2 and H3K79me1. However, previous cytological analyses (Ref. 23) go in the opposite direction because Dot1-dependent H3K79 methylation appears to prevent Pch2 mislocalization to euchromatin. That study shows that Pch2 mislocalizes outside the rDNA in dot1 and H3K79 mutants. The authors have opted for not directly analyzing transcription-dependent Pch2 binding in dot1 (and sir2) mutants arguing time restrictions. Although genome-wide ChIP-seq studies could take more time, at least ChIP-qPCR analyses at the same selected target genes explored in other experiments could have been done.

Minor points:

Line 245: Supplementary Figure 3E should be 4E

Line 270: This (delete “is”) observation is in agreement…

Lines 390-392 and Supplementary Figure 8G, 8H. Has been the nucleolus included in the quantification of Hop1 “chromosomal” levels? Ref 39 shows that the absence of Pch2 in the nucleolus in orc1 mutants leads to Hop1 mistargeting to the rDNA region. Please, mark the nucleolus in the spread nucleus displayed in Fig S8G to show that Hop1 is mistargeted to the rDNA in orc1-161. The nucleolar area must be eliminated for accurate Hop1 chromosomal quantification.

Line 407: Supplementary Figure 7A-C is erroneously cited here.

Reviewer #3: Please see attachment

**Have all data underlying the figures and results presented in the manuscript been provided?**

Reviewer #1: Yes

Reviewer #2: Yes

Reviewer #3: None

PLOS authors have the option to publish the peer review history of their article (what does this mean?). If published, this will include your full peer review and any attached files.

Reviewer #1: No

Reviewer #2: No

Reviewer #3: No

---

## [Editor Report · Decision Letter 2]

30 May 2020

Dear Gerben,

Thank you very much for submitting your Research Article entitled 'Active transcription and Orc1 drive chromatin association of the AAA+ ATPase Pch2 during meiotic G2/prophase' to PLOS Genetics, and thank you for the revisions that you made in response to the reviewers' comments. I am just about ready to proceed with this manuscript, but in reviewing the previous version I slipped up and did not pay attention to the references section, which needs revision to conform to PLOS Genetics style. Unfortunately, because of the way article production works, it is difficult to make substantial changes after a paper has been accepted, and I have been advised in the past that it is better to return the manuscript so that these changes can be made before starting production. I apologize for the inconvenience to you, because I know how much time doing a resubmission can take.

If you will refer to Author instructions (journals.plos.org/plosgenetics/s/submission-guidelines), you will find that the recommended format is as follows, in particular using journal abbreviations rather than full journal names:

Hou WR, Hou YL, Wu GF, Song Y, Su XL, Sun B, et al. cDNA, genomic sequence cloning and overexpression of ribosomal protein gene L9 (rpL9) of the giant panda (Ailuropoda melanoleuca). Genet Mol Res. 2011;10: 1576-1588.

Devaraju P, Gulati R, Antony PT, Mithun CB, Negi VS. Susceptibility to SLE in South Indian Tamils may be influenced by genetic selection pressure on TLR2 and TLR9 genes. Mol Immunol. 2014 Nov 22. pii: S0161-5890(14)00313-7. doi: 10.1016/j.molimm.2014.11.005.

Note: A DOI number for the full-text article is acceptable as an alternative to or in addition to traditional volume and page numbers. When providing a DOI, adhere to the format in the example above with both the label and full DOI included at the end of the reference (doi: 10.1016/j.molimm.2014.11.005). Do not provide a shortened DOI or the URL.

Also, if you used Endnote, you can find a link to the appropriate style in the references section of the instructions. Please also make sure that all titles are in sentence case (like this sentence); often when articles are downloaded from journals their titles have Every Word Capitalized and this must be corrected by editing the reference manager entry or the references section. 

Finally, as a matter of genetics style, please make sure that all gene and species names are italicized--you got it correct for most but I found at least one (ref 50) where it wasn't, and may have missed a few others.

We therefore ask you to modify the manuscript according to the review recommendations before we can consider your manuscript for acceptance. Your revisions should address the specific points made by each reviewer.

The rest of this letter is boilerplate--thanks for your interesting article, and I hope to receive the final version soon.

[LINK]

Yours sincerely,

Michael Lichten, Ph.D.

Associate Editor

PLOS Genetics

Gregory P. Copenhaver

Editor-in-Chief

PLOS Genetics

---

## [Editor Report · Decision Letter 3]

3 Jun 2020

Dear Gerben,

We are pleased to inform you that your manuscript entitled "Active transcription and Orc1 drive chromatin association of the AAA+ ATPase Pch2 during meiotic G2/prophase" has been editorially accepted for publication in PLOS Genetics. Congratulations!

Yours sincerely,

Michael Lichten, Ph.D.

Associate Editor

PLOS Genetics

Gregory P. Copenhaver

Editor-in-Chief

PLOS Genetics

Comments from the reviewers (if applicable):

**Data Deposition**

http://datadryad.org/submit?journalID=pgenetics&manu=PGENETICS-D-19-01662R3

**Press Queries**

---

## [Editor Report · Acceptance letter]

15 Jun 2020

PGENETICS-D-19-01662R3 

Active transcription and Orc1 drive chromatin association of the AAA+ ATPase Pch2 during meiotic G2/prophase 

Dear Dr Vader, 

We are pleased to inform you that your manuscript entitled "Active transcription and Orc1 drive chromatin association of the AAA+ ATPase Pch2 during meiotic G2/prophase" has been formally accepted for publication in PLOS Genetics! Your manuscript is now with our production department and you will be notified of the publication date in due course.

With kind regards,

Jason Norris

PLOS Genetics

On behalf of:
